# Microinterfaces in biopolymer-based bicontinuous hydrogels guide rapid 3D cell migration

Karen L. Xu [1,2,3,4], Nikolas Di Caprio[1,2], Hooman Fallahi [5], Mohammad Dehghany[2,6], Matthew D. Davidson[2,7,8], Lorielle Laforest[3,4], Brian C. H. Cheung[9], Yuqi Zhang[1,3,4], Mingming Wu [9], Vivek Shenoy [2,6], Lin Han [5], Robert L. Mauck [1,2,3,4] ✉ & Jason A. Burdick [1,2,7,8] ✉

Cell migration is critical for tissue development and regeneration but requires extracellular environments that are conducive to motion. Cells may actively generate migratory routes in vivo by degrading or remodeling their environments or instead utilize existing extracellular matrix microstructures or microtracks as innate pathways for migration. While hydrogels in general are valuable tools for probing the extracellular regulators of 3-dimensional migration, few recapitulate these natural migration paths. Here, we develop a biopolymer-based bicontinuous hydrogel system that comprises a covalent hydrogel of enzymatically crosslinked gelatin and a physical hydrogel of guest and host moieties bonded to hyaluronic acid. Bicontinuous hydrogels form through controlled solution immiscibility, and their continuous subdomains and high micro-interfacial surface area enable rapid 3D migration, particularly when compared to homogeneous hydrogels. Migratory behavior is mesenchymal in nature and regulated by biochemical and biophysical signals from the hydrogel, which is shown across various cell types and physiologically relevant contexts (e.g., cell spheroids, ex vivo tissues, in vivo tissues). Our findings introduce a design that leverages important local interfaces to guide rapid cell migration.

Cell migration is essential for many biological processes including tissue development[1], wound healing[2], and regeneration[3]. Migration is guided by a number of extracellular signals, including biochemical gradients (chemotaxis or haptotaxis) and biophysical cues (durotaxis or topotaxis)[4]. While these principles are well established in planar (2D) contexts, most cells migrate in vivo in a 3D context. This added dimension is a critical consideration, given the increased complexity in the presentation of physical features (e.g., matrix density, porosity) in 3D when compared to cells migrating along a surface in 2D[5]. Engineered hydrogels are commonly used to model extracellular environments to study cell migration in 3D and have demonstrated that bulk features such as degradability and/or viscoplasticity can influence

[1]Department of Bioengineering, University of Pennsylvania, Philadelphia, PA 19104, USA. [2]Center for Engineering Mechanobiology, University of Pennsylvania, Philadelphia, PA 19104, USA. [3]McKay Orthopaedic Research Laboratory, Department of Orthopaedic Surgery, Perelman School of Medicine, University of Pennsylvania, Philadelphia, PA 19104, USA. [4]Translational Musculoskeletal Research Center, Corporal Michael J. Crescenz VA Medical Center, Philadelphia, PA 19104, USA. [5]School of Biomedical Engineering, Science and Health Systems, Drexel University, Philadelphia 19104 PA, USA. [6]Department of Materials Science and Engineering, University of Pennsylvania, Philadelphia, PA 19104, USA. [7]BioFrontiers Institute, University of Colorado Boulder, Boulder, CO 80303, USA. [8]Department of Chemical and Biological Engineering, University of Colorado Boulder, Boulder, CO 80303, USA. [9]Department of Biological and Environmental Engineering, Cornell University, Ithaca, NY 14850, USA. ✉e-mail: lemauck@pennmedicine.upenn.edu; jason.burdick@colorado.edu

single cell migration[6–8]. Despite these important findings, studies with monolithic hydrogels have largely neglected the impact of 3D extracellular matrix (ECM) microstructures on cell migration[9,10].

Most connective tissues consist of a fibrillar ECM (e.g., collagen) that is interwoven with proteoglycans. Migration through such dense networks may occur as the result of proteases (e.g., matrix metalloproteases), where cell-mediated matrix degradation generates migratory paths through the ECM[11]. The ECM microstructure may also impact and enable cell migration, such as with the stiffening and linearization of fibrillar ECM components in the tumor microenvironment, which promotes directional migration towards vasculature and results in cellular metastasis[12]. Likewise, in the meniscus, a highly aligned dense connective tissue in the knee, migration through the dense ECM occurs preferentially in directions that are defined by the internal architecture of the tissue[13,14]. Such microtracks – pathways either inherently present in tissues (e.g., between collagen bundles, or along nerve strands or small blood vessels) or that develop during ECM remodeling - provide microinterfaces that allow for rapid cell migration in 3D[15,16]. These microtracks often guide initial migration direction, with cells conforming to microtrack topology. In vitro studies introducing microtracks into hydrogels (via laser ablation[17,18] and microfabrication[19,20]) show that migration along preformed pathways is energetically favorable and faster than migration through isotropic gels. While in many diseases (e.g., cancer), it would be desirable to avoid microtracks and resulting cell migration (metastasis), such features could also be harnessed to enable rapid cell migration for applications in tissue repair.

To exploit these features and promote cell migration along such microinterfaces in 3D, we focus here on the use of bicontinuous materials as a unique platform for investigation. Bicontinuous materials consist of two independent domains that are each continuous but distinct within the macrostructure[21–23]. Due to the combination of different domains, bicontinuous structures may have superior material properties than each composition alone. These unique structures also produce high interfacial surface areas, which have previously been leveraged to generate electrical conductors[24,25] as well as tough, fatigue-resistant materials with multiscale resistance to crack propagation[26–28]. Such a system may be useful to guide cell migration as well, given that individual domains can be designed to present migratory cues to cells along a high surface area in 3D space.

Here, we introduce a biopolymer-based bicontinuous hydrogel, in which the interactions of distinct biopolymers (i.e., gelatin and hyaluronic acid) that are abundant in the ECM engender extensive 3D interfaces with microscale topologic features that influence cell migration. These bicontinuous structures are formed via controlled miscibility during hydrogel formation, with domain length scales defined by the material formulation. These controllable microstructural features in turn regulate the extent of cell migration from cell aggregates and tissues. Meniscal cells and tissue are used as platforms to illustrate this phenomena, due to the prevalence of meniscal injuries and the need for new approaches for repair[29,30]. Ultimately, these data support that the bicontinuous structure may be useful to program migratory cell behaviors.

## Results
### Fabrication of a biopolymer-based bicontinuous hydrogel
To fabricate a cell-compatible biopolymer-based bicontinuous hydrogel, we developed an approach that limits the mixing of hydrogel precursor components to achieve a hydrogel with discrete 3D material domains. Specifically, we combined the components of (i) a guest-host (GH) hyaluronic acid (HA)-based physical network (where near instantaneous dynamic complexes form between cyclodextrin (host, CD-HA) and adamantane (guest, AD-HA) moieties when independently conjugated onto HA [see supplemental NMR, Supplementary Fig. 1]) and (ii) a covalently crosslinked gelatin network (that forms through

the enzymatic crosslinking of gelatin with transglutaminase). Hydrogels were prepared by mixing two separate solutions, the first of which contained AD-HA and gelatin and the second of which contained CD-HA and transglutaminase (Fig. 1a). These solutions were added to a blunt-ended syringe and mixed with circular revolutions until the structure stabilized (>10 circular revolutions, Supplementary Fig. 2a, Supplementary Movie 1). When combined, the interactions between AD and CD immediately limited miscibility of the solutions. Over the ensuing tens of minutes, some dynamic material changes were observed (Supplementary Movie 2). However, the enzymatic activity of the transglutaminase stabilized the discrete hydrogel domains after several hours. To explore the emergence of discrete domains, GH concentration was varied across hydrogel formulations while maintaining consistent gelatin (5 wt% gelatin) and transglutaminase (1 U/mL) concentrations. Fluorescent visualization (via the inclusion of fluorescent gelatin) confirmed the uniformity of gelatin-only hydrogels (0% GH) and the emergence of a tortuous 3D structure characteristic of a bicontinuous material as the concentration of GH increased (Fig. 1b). These hydrogels contained distinct gelatin-rich (GR) and gelatin-poor (GP) domains. At lower GH concentrations (0.5–1%), the GP domains were generally spherical in morphology and were sparsely distributed throughout a continuous GR domain. These GP domains increased in size, with some connectivity between domains emerging at slightly higher GH concentrations (1–2%). At the highest GH concentrations (3–4%), these GP domains formed an interconnected bicontinuous structure.

To better explore the chemical composition in the bicontinuous hydrogels, individual components were fluorescently labeled prior to mixing and gelation. We observed with a 3% GH formulation (Supplementary Fig. 2b) that polymers colocalize based on the original solutions used in hydrogel formation (Fig. 1a). That is, gelatin and AD-HA (guest) were the primary components of the GR domain, while CD-HA (host) was the main constituent of the GP domain. Given that transglutaminase (which was originally in the CD-HA solution) must interact with gelatin to crosslink that domain (Fig. 1c), all components of the two solutions are likely diffusing and interacting with each other within and across the domains, to varying extents. These findings suggest that the strong physical interactions between the components of the two solutions result in an immiscibility that stabilizes the interfaces into bicontinuous structures, where each domain consists of a unique hydrogel composition.

To further evaluate these hydrogel structures, three formulations were identified (5 wt% gelatin, with 0, 1, 3% GH) to represent either uniform hydrogels (0% GH) or hydrogels with varying levels of bicontinuity (1, 3% GH). Rheological profiles after mixing showed rapid gelation (G' > G") and a plateau in moduli over hours (~150 min) (Fig. 1c), with little influence of the enzyme concentration on hydrogel crosslinking time (slight reduction in gelation time with increased enzyme concentrations) and hydrogel properties (Supplementary Fig. 3). A slight increase in loss modulus was observed over time across increasing GH concentrations; yet, G" in the 0% GH hydrogel remained much lower than that observed when GH was incorporated, likely due to the absence of dynamic bonds from the GH network. After 150 min, the hydrogels stabilized structurally and mechanically (Fig. 1d, Supplementary Movie 2). The rheological assessment showed that the formed hydrogels exhibited viscoelastic properties that were dependent on the formulation (Fig. 1e). Specifically, with a consistent gelatin concentration (5 wt%) and enzyme concentration (1 U/mL), increasing the GH concentration resulted in consistent oscillatory elastic moduli (G'), but increased loss moduli (G") and tan (δ). Within the linear viscoelastic regime, the higher GH concentrations showed increased frequency dependency, with the 3% formulation having a higher viscous contribution compared to the 0 and 1% formulations (Supplementary Fig. 4a–c). 1% and 3% GH formulations also underwent stress relaxation (Supplementary Fig. 4d) and showed some viscoplasticity,

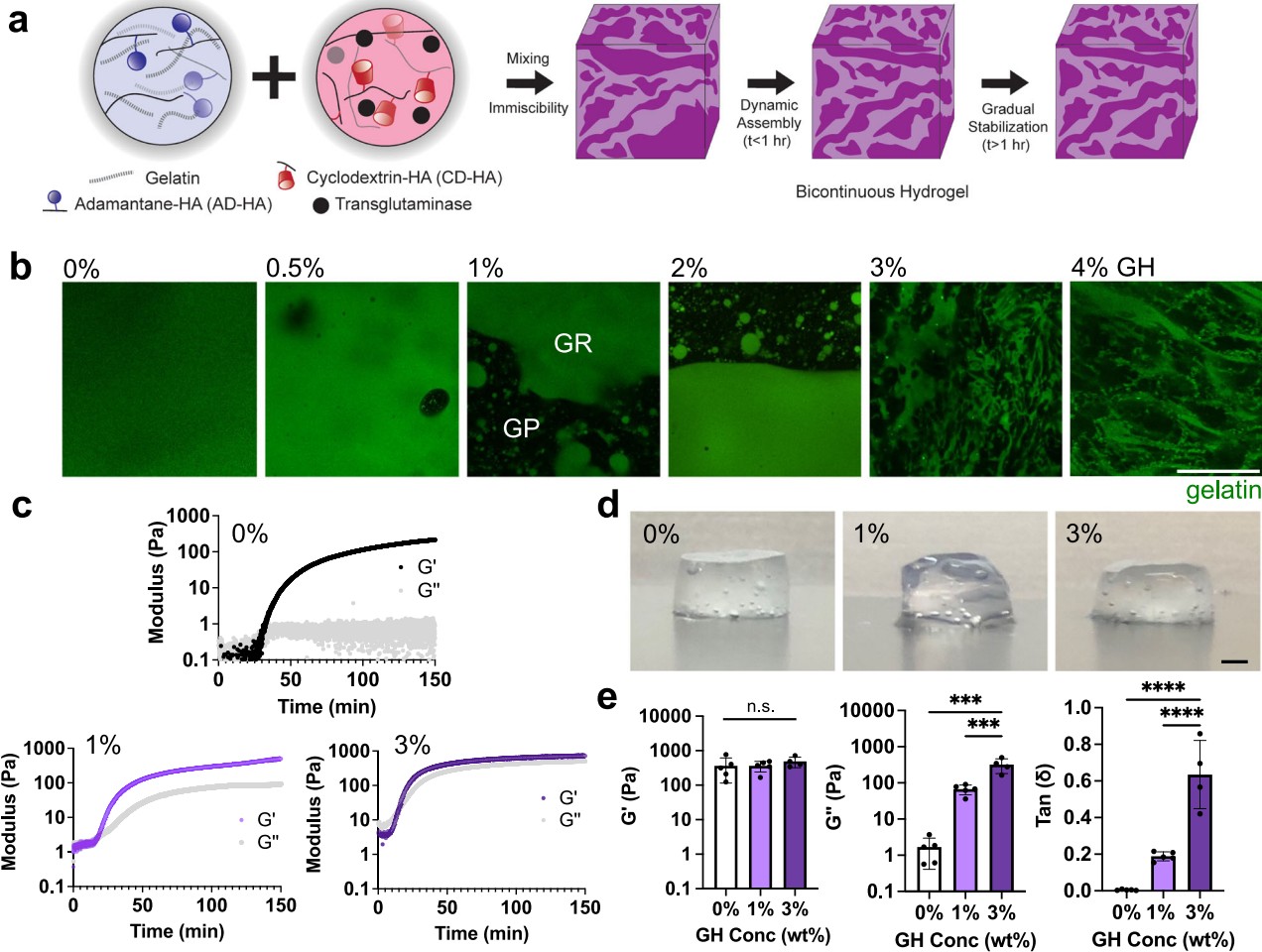

**Fig. 1 | Fabrication of biopolymer-based bicontinuous hydrogels. a** Schematic of bicontinuous hydrogel fabrication through the mixing of components to form gelatin (crosslinked with transglutaminase) and hyaluronic acid (crosslinked with adamantane (AD-HA) and cyclodextrin (CD-HA) guest-host (GH) complexes) hydrogels. The bicontinuous structure evolves through initial network immiscibility and then dynamic assembly and stabilization over hours. **b** Representative confocal fluorescent images of hydrogels (gelatin: green) where increasing GH content results in gelatin-rich (GR) and gelatin-poor (GP) domains. Scale bar = 50 μm.

**c** Representative rheological time sweeps (storage (G′) and loss (G″) modulus, 1 Hz, 1% strain) illustrating the kinetics of gelation. **d** Representative macroscopic images (Scale bar = 1 mm), and **e** quantified G′, G″, and tan (δ) after gelation (150 min) for hydrogels containing various GH content (0, 1, 3 wt%). $n = 4$ (0,1%) or 5 (3%) hydrogels per condition. G′: n.s. indicates no statistical significance; G″: 0% vs. 1% ***$p = 0.0001$; 1% vs. 3% ***$p = 0.0009$; tan(δ): 0% vs. 3%, 1% vs. 3% ****$p \le 0.0001$; one-way ANOVA with Tukey post hoc. Data are mean ± s.d. Source data for (**c**, **e**) provided as a source data file.

as illustrated by residual strain after creep-recovery assays (Supplementary Fig. 4e)[31]. Importantly, hydrogels fabricated without the enzymatic crosslinker did not form robust hydrogels, remaining as a solution in the case of gelatin alone hydrogels or as a soft, physical network when gelatin and GH were combined (Supplementary Fig. 4f). Additionally, a comparison of the 0% GH hydrogel (gelatin-only) and GH-only hydrogels highlights the increased viscoelastic properties of the physical GH hydrogel when compared to the covalent gelatin hydrogel (Supplementary Fig. 4g, h).

Confocal microscopy was used to further characterize morphological features of the hydrogel formulations and to confirm their bicontinuous structure. Confocal sections of hydrogels (750 × 750 μm region in 0, 1, 3% GH) clearly showed heterogeneous structures with high coefficients of variation when GH was included (Supplementary Fig. 5a, b). Notably, both components of the GH hydrogel were required to achieve these structures in hydrogels - if only one component of the material (AD-HA or CD-HA) was included, hydrogels exhibited uniform fluorescence and were similar in appearance to the 0% GH hydrogels (Supplementary Fig. 5c). 3D reconstructions of the 0, 1, and 3% GH hydrogels demonstrated that an increase in GH content led to emergent bicontinuity, decreased

volume fractions of GR domains, and increased volume fractions of GP domains (Fig. 2a, b, Supplementary Movie 3). While hydrogel precursor solutions were mixed at a 50/50 volume ratio, the volume fraction of each domain was not consistently 50%, suggesting that the separation of GR and GP domains is not purely due to immiscibility, but rather also relies on diffusion between components that depends on the GH content. Since the tortuous intercalation of the domains was clear, we next sought to quantify their connectivity using an object-based analysis similar to the flood-fill pathway algorithm, in which the volume of the largest discrete object within each discrete domain was normalized to the total volume of the domain[32]. Here, the presence of GH led to connected structures in both the GR and GP domains, as >90% and often nearly 100% of the domain was connected as a continuous object (Fig. 2c). Furthermore, as with other bicontinuous materials[26], these hydrogels exhibited high interfacial surface areas between the domains, with the 3% GH hydrogel having the highest surface area when compared to the 0 and 1% GH hydrogels (Fig. 2d). These findings suggest that increasing GH content increases the domain interpenetration and connectivity. Importantly, the hydrogels were also structurally and chemically stable for at least a week (Supplementary Fig. 6).

To determine if the observed heterogeneity in structure (Supplementary Fig. 7a) correlated with mechanical heterogeneity at the microscale, atomic force microscopy was used to generate force maps (105 × 105 μm) (Fig. 2e). Average moduli across samples changed with formulation (0% had a higher modulus than 3% hydrogels) and, similar to the structural assessment, mechanical microheterogeneity increased as a function of increasing GH (as quantified by an increase in the coefficient of variation within a force map, Fig. 2f). Based on these

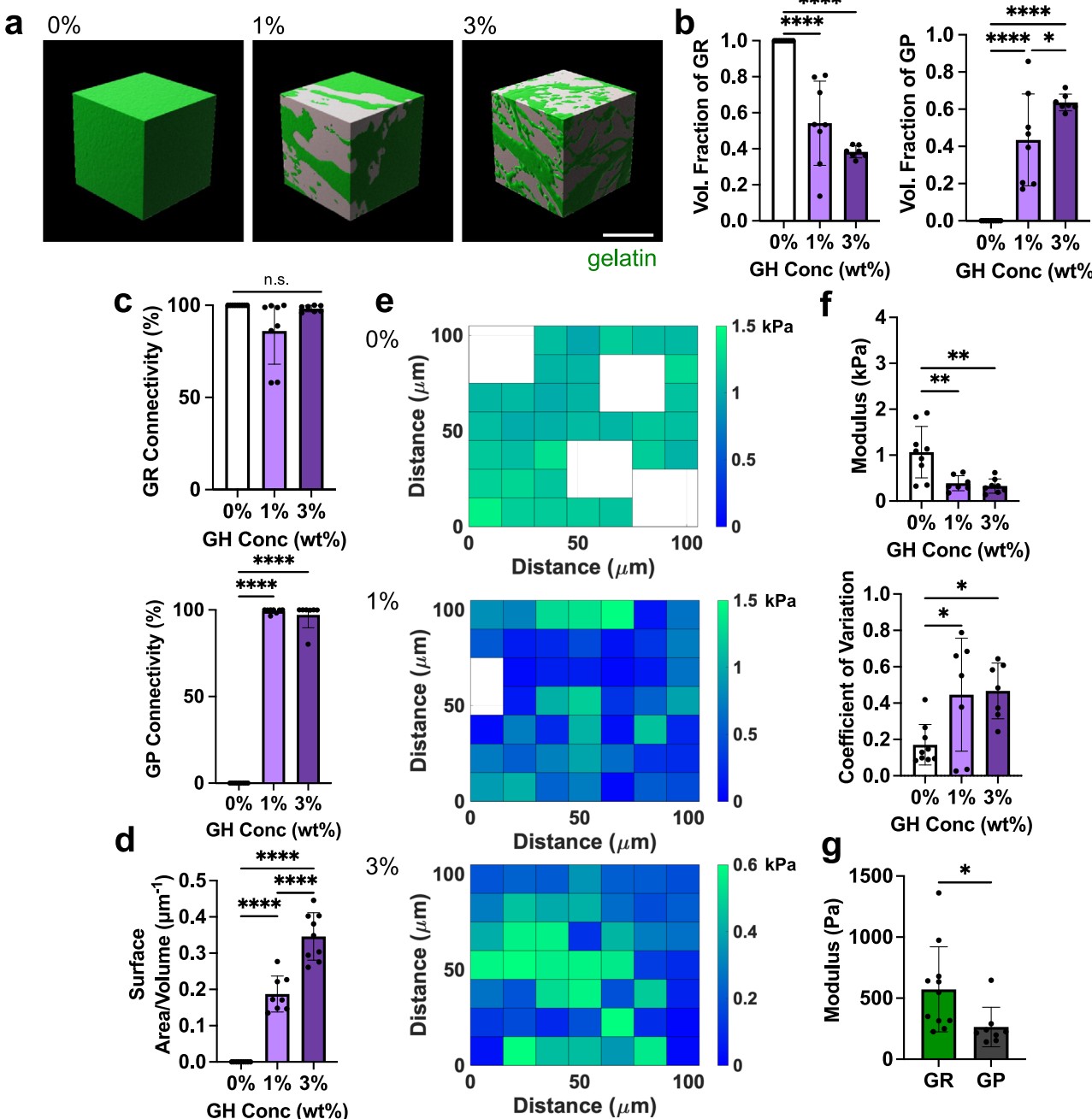

**Fig. 2 | Bicontinuous hydrogel structural and mechanical characterization.**
**a** Representative 3D reconstructions of hydrogels containing various GH content (0, 1, 3 wt%), revealing bicontinuous structures in 1 and 3 wt% hydrogels (gelatin-rich (GR, green) and gelatin-poor (GP, gray) domains). Scale bar = 50 μm. **b** Fraction of total volume occupied by GR (left panel) and GP (right panel) domains within hydrogels. $n = 9$ (0%), 8 (1%), or 7 (3%) regions across 3 hydrogels per condition. Left panel: 0% vs. 1%, 1% vs. 3% ****$p ≤ 0.0001$. Right panel: 0% vs. 1%, 0% vs. 3% ****$p ≤ 0.0001$; 1% vs. 3% *$p = 0.0353$; one-way ANOVA with Tukey post hoc. **c** Quantification of connectivity (defined by percent of total volume occupied by largest individual domain) of GR (top) and GP (bottom) domains. $n = 9$ (0%), 8 (1%), or 7 (3%) regions across 3 hydrogels per condition. Upper panel: n.s. indicates no statistical significance. Lower panel: 0% vs. 1%, 0% vs. 3% ****$p ≤ 0.0001$; one-way ANOVA with Tukey post hoc. **d** Quantification of interfacial surface area normalized to sample volume across hydrogels containing various GH content (0, 1, 3 wt%). $n = 9$ (0, 3%) or 8 (1%) regions across 3 hydrogels per condition. 0 % vs. 1%, 0% vs. 3%, 1% vs. 3% ****$p ≤ 0.0001$; one-way ANOVA with Tukey post hoc. **e** Representative elastic modulus maps determined with AFM-nanoindentation (empty nodes denote removed outliers). **f** Average elastic moduli (top) and coefficient of variation (CV, bottom) across AFM profiles. $n = 9$ (0%), 7 (1%), 8 (3%) grids across 3-4 hydrogels per condition. Upper panel: 0% vs. 1% **$p = 0.0041$; 0% vs. 3% **$p = 0.0014$. Lower panel: 0% vs. 1% *$p = 0.0354$; 0% vs. 3% *$p = 0.0228$; one-way ANOVA with Tukey post hoc. **g** Quantification of averaged elastic moduli of GR and GP domains across probed grids for 1 wt% GH hydrogel using AFM based on differential fluorescent labeling (individual data points in Supplementary Fig. 7d). $n = 8$ (GP) or 11 (GR) grids across 5 hydrogels. *$p = 0.0333$; Two-tailed unpaired students t-test. Data are mean ± s.d. Source data for (**b**–**g**) provided as a source data file.

observations, we next performed AFM-nanoindentation to measure the distinct mechanical properties of the GR and GP domains, using fluorescence hydrogel image maps to differentially probe discrete regions (Supplementary Fig. 7b–d). As illustrated with the 1% GH hydrogel, GR domains had higher mechanical properties compared to GP domains (Fig. 2g). This is consistent with our understanding that the stiffer covalent gelatin hydrogel is enriched in the GR domains, whereas the primarily physical networks in the GP domains exhibit a lower modulus. Importantly, the GP domains had non-zero mechanical properties, further confirming the presence of a weak hydrogel (rather than open pores) that contributes to the overall mechanical behavior of the material. Furthermore, mechanical properties were equivalent on external (i.e., surface) and internal surfaces of the hydrogel (Supplementary Fig. 7e), confirming that surface-level mechanics are representative of internal properties as well. Overall, this material characterization indicates that the 0, 1, and 3% hydrogels possessed a range of structural and physical features that can be used to explore the impact of hydrogel bicontinuity on cell migration.

## Bicontinuous hydrogels support rapid cell migration

Given the importance of cell migration in multiple physiologic contexts (e.g., meniscus), a number of studies have employed spheroid outgrowth assays to query how varying extracellular environments influence cell motility[33]. To explore this in our bicontinuous hydrogels, meniscal fibrochondrocyte (MFC) spheroids (1000 cells/spheroid) were embedded within hydrogels of varying GH concentration (Fig. 3a). Direct observation of migration in real time showed minimal migration in 0% GH hydrogels, but a marked outflow of cells in the 1% and 3% hydrogels (Supplementary Movie 4). Notably, in the 3% GH hydrogels, cells appeared to utilize the interfaces between the GR and GP domains as paths for outward growth (Fig. 3b). This is consistent with previous studies showing that cells spread along interfaces between different densities of collagen[34]. Interestingly, cells did not enter into larger length-scale GR domains, which would likely require enzymatic degradation. Similarly, cells traversed GP domains only when they were interspersed with abundant GR domains. Decreasing enzymatic crosslinker concentration similarly supported cell migration along interfaces in the 3% bicontinuous hydrogel (Supplementary Fig. 8). These findings suggest that the high surface area and microinterfaces of the bicontinuous hydrogels may provide favorable paths for cell migration.

Previous studies have identified a number of extracellular characteristics, including porosity, degradation and viscoelasticity that promote cell migration in 3D compared to substrates that lack these biophysical cues[31,35,36]. To explore if this phenomenon was also true with the addition of microinterfaces, we next measured cell migration speed and distance. Total radius of outgrowth over 3 days was highest in the 3% GH hydrogel (Fig. 3c, d). Similarly, average speed increased in the presence of microinterfaces (Fig. 3e, f, Supplementary Movie 5) and was comparable between bicontinuous hydrogels (1 and 3%). For both measures, the 0% GH hydrogel supported minimal outgrowth over this time. This further supports that the microinterfaces that arise in bicontinuous hydrogels enable migration in 3D.

To further characterize cell migration trajectories in bicontinuous hydrogels, we utilized the anisotropic persistent random walk (APRW) model in our 3% GH hydrogel, given that this hydrogel formulation was most conducive to cell migration (Fig. 3d)[37]. According to the APRW model, 3D cell migration can be described as persistent random walks along two axes, the primary ($\vec{\mathbf{p}}$) and non-primary ($\mathbf{n}\vec{\mathbf{p}}$) axes of migration[37]. We first showed that population-averaged migration speed increases after ~30 h, which is likely after cells have fully left the spheroid body (Supplementary Fig. 9a). After this time point, the population-averaged cell speed was nearly independent of time, making it possible to use the APRW model during this period of migration. We further assumed that cell proliferation during this

observation period was minimal. We calculated the population-averaged mean squared displacements (MSD) of cells (Fig. 3g), which demonstrated an exponent $\alpha \sim 1$ (measured from a fit of $MSD \sim t^{\alpha}$), indicating that cell motility in our hydrogel system incurs random (Brownian) motion and may not be due to crowding (sub-diffusion) or targeted cell movement (super-diffusion)[38,39]. Probability density functions of cell angular displacements, $d\theta$, during migration (Supplementary Fig. 9b) further showed that the probability of observing small and large (around $180^{0}$) $d\theta$ values is persistently high during cell migration, which contrasts with the isotropic random walk of cells migrating on 2D substrates where the density function becomes flat over time[37]. This suggests that the bicontinuous structure offers preferential migration paths (formed from domain interfaces) that render the cell migration anisotropic (directional).

Next, to determine the primary migration directions, we performed singular value decomposition (SVD) of individual cell velocities and realigned our coordinate system with the obtained primary ($\vec{\mathbf{p}}$) and non-primary ($\mathbf{n}\vec{\mathbf{p}}$) axes of migration. Using this rotated system, we calculated the velocity magnitude profiles of cells at different orientations relative to $\vec{\mathbf{p}}$ (Fig. 3h) and observed that cells display higher velocities along their primary axis of migration (Supplementary Fig. 9c). Furthermore, we extracted the MSDs along the two migration axes of individual cells and fitted them to the APRW model (see Methods) to obtain the population-averaged cell speeds and persistent times, which are 38.47 μm/hr and 0.56 hr (Supplementary Fig. 9c, d) along $\vec{\mathbf{p}}$. Accordingly, we calculated cell diffusivities along the $\vec{\mathbf{p}}$ and $\mathbf{n}\vec{\mathbf{p}}$ axes (Supplementary Fig. 9e) and obtained the total cell diffusivity in our 3% GH hydrogel as 147 μm²/hr. We also found the anisotropy index ($\phi$), the ratio of the diffusivities, to be 1.9 (Supplementary Fig. 9f), indicating that cell movement in our bicontinuous networks exhibits anisotropy. Interestingly, this anisotropic index is lower than that found in collagen gels of varying densities, suggesting that our hydrogels are less susceptible to the remodeling and force-exertion that results in increasing anisotropic behavior in those systems[37], but may still promote anisotropic behaviors based on the structure of domain interfaces. Finally, we verified the accuracy of the APRW model by simulating cell migration trajectories based on the experimental values obtained for the persistent time and speed and observed good agreement of the MSD and velocity magnitude profiles between the experimental and simulated results (Fig. 3g, h). Together, these data demonstrate that cell migration in our bicontinuous hydrogels follows similar trends to previously observed 3D cell migration[37,40].

Importantly, the trends in migration were consistent across multiple cell types (outgrowth of 386 μm for MFCs and 413 μm in mesenchymal stromal cells (MSCs) in 3% GH hydrogel after 3 days), suggesting a general cell-type independent response to the hydrogels (Supplementary Figs. 10, 11a, b). Additional measurements of proliferation and matrix production showed low levels of proliferation and progressive deposition of nascent matrix as cells migrated away from the spheroid and through the hydrogel (Supplementary Figs. 10, 11c, d). This indicates that not only do the bicontinuous hydrogels enable migration, but they also support cell behaviors that are potentially important for tissue regeneration[41]. Notably, the addition of soluble HA (rather than HA in the GH context) resulted in a homogenous hydrogel in which cells had minimal migration (similar to a 0% GH hydrogel), indicating that the presence of HA alone is insufficient to drive migration (Supplementary Fig. 12a–c). Taken together, these findings show that bicontinuous hydrogels provide routes for the rapid migration of cells in 3D[31].

## Biophysical and biochemical determinants of cell migration in bicontinuous hydrogels

Having established that the bicontinuous structure promotes cell migration, we next sought to determine precisely how cells navigate through this network. Many studies have noted a relationship between

migration speed and traction forces exerted by cells on 2D substrates (by tracking bead displacements)[42]. Others have explored cell generated forces that are exerted within elastic, isotropic 3D matrices[43,44]. Building on these approaches, we tracked bead displacement (as a proxy for traction force) to explore how cells exert forces on and interact with the bicontinuous hydrogels during migration[45]. Matrix-

displacement maps revealed that cells actively exert inward tractions (pull on the hydrogel) at sites of cell protrusion directly in front of the migration path (Fig. 4a–d). Cells continue to displace the hydrogel in the direction of migration as they traverse through a region, and then release this tension on the hydrogel after they have passed. This suggests that cells are actively pulling on their trailing edge prior to release

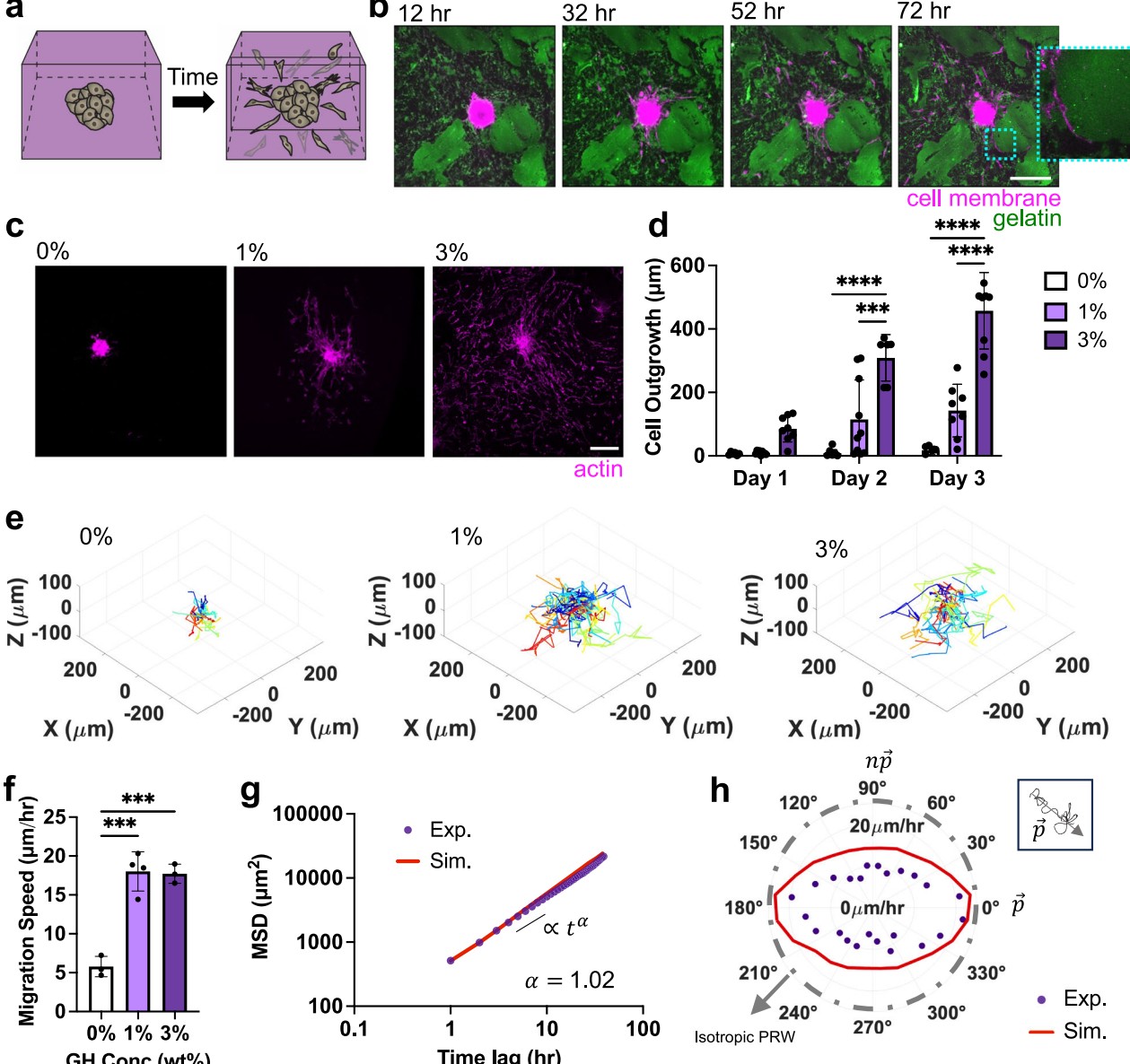

**Fig. 3 | Bicontinuous hydrogels introduce micro-interfaces that support rapid cell migration. a** Schematic of meniscal fibrochondrocyte (MFC) spheroid embedded within hydrogels and subsequent MFC migration over time. **b** Representative Z sections of a MFC spheroid (magenta) embedded within a 3 wt% GH bicontinuous hydrogel (gelatin-rich (GR) domain: green) over time. Inset (dotted blue border) is a single Z-section zoom-in highlighting cells along GR domain. Scale bar = 200 μm. **c** Representative maximum projection images (actin: magenta, Scale bar = 200 μm) at 3 days and **d** quantified outgrowth of MFCs over 3 days for spheroids embedded in hydrogels containing various GH content (0, 1, 3 wt%). *n* = 6 (Day 3:0%, Day 2:3%), 8 (Day 1:0%, Day 3:1%, Day 1: 3%), 9 (Day 2:0%, Day 1:1%, Day 3:3%) or 10 spheroids (Day 2:1%) from 2 biologically independent experiments. Day 2:0% vs. 3% ****$p \leq 0.0001$; Day 2:1% vs. 3% ****$p = 0.0001$; Day 3:0% vs. 3%, Day 3:1% vs. 3%, ****$p \leq 0.0001$; two-way ANOVA with Tukey post hoc. **e** Representative 3D migration tracks over 3 days, with track starting from plot origin (each cell track denoted with different color, *n* = 21–26 representative tracks

per condition) and **f** quantified migration speeds of MFCs from spheroids embedded in hydrogels containing various GH content (0, 1, 3 wt%). *n* = 3 (0,3%) or 4 (1%) spheroids per condition from 1 biologically independent experiment. 0% vs. 1% ***$p = 0.002$; 0% vs. 3% ***$p = 0.003$; one-way ANOVA with Tukey post hoc. **g** Representative population-averaged mean squared displacement of cells over different time lags (starting from 30 h after spheroid embedding) computed from experimental (Exp) data and simulated (Sim) trajectories with exponential α, fitted based on MSD ∝ $t^\alpha$ in 3% GH hydrogels. *n* = 3 spheroids across 1 biologically independent experiment. **h** Representative experimental (Exp) and simulated (Sim) velocity magnitude profiles as a function of varying orientations relative to the primary axis $\vec{p}$ for 3% GH hydrogel across cells from one spheroid, in which purple circles denote the average magnitude of cell velocity in a specified direction. Inset shows that the primary axis is determined by the vector of cell trajectory over the period of migration. Data are mean ± s.d. Source data for (**d**, **f**–**h**) provided as a source data file.

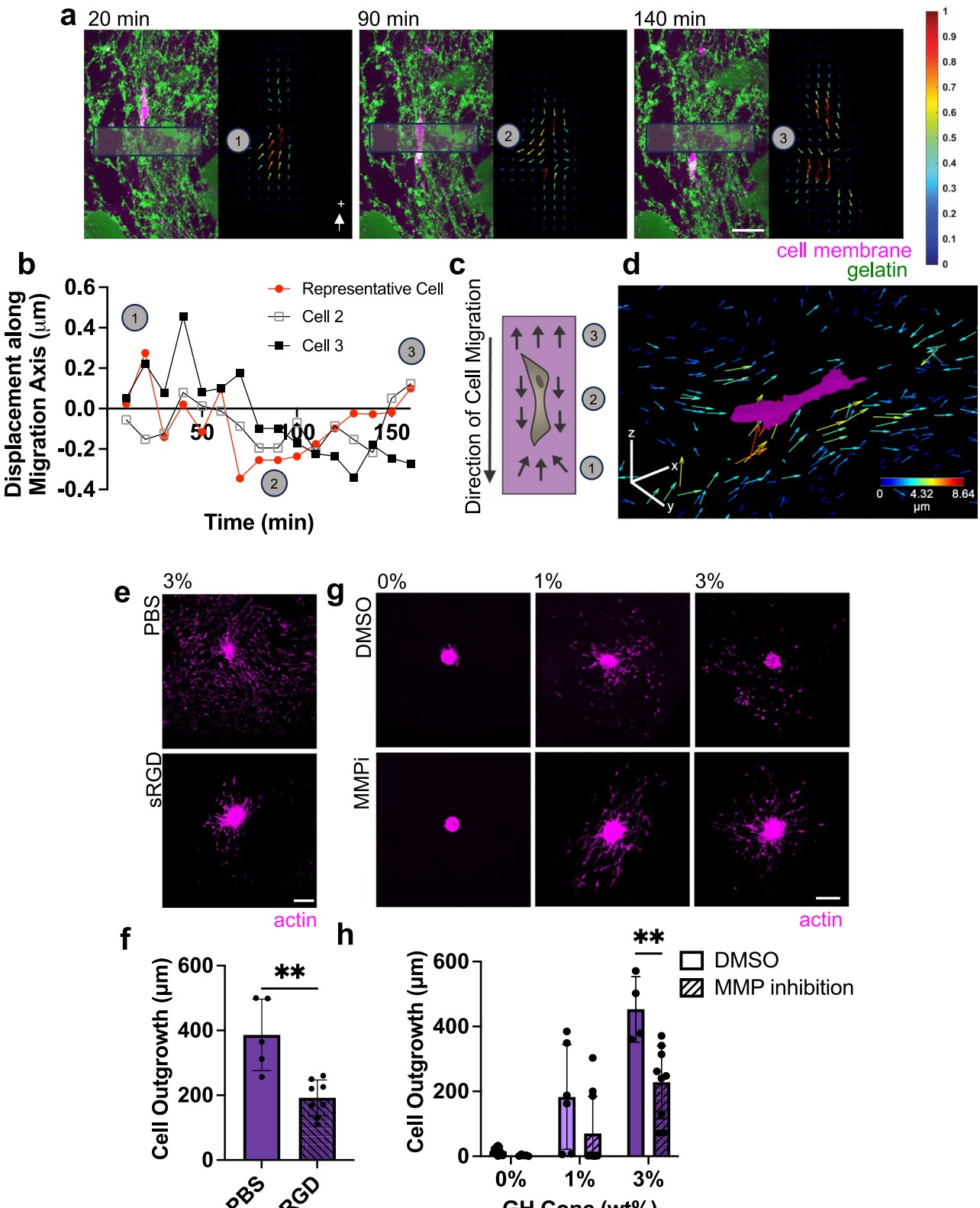

(Supplementary Movie 6). These cell migration strategies are similar to those of cells migrating through isotropic fibrous collagen-based networks[46,47] and suggest that cells rely on myosin-based contractility and mesenchymal mechanisms of migration (which is supported by the extracellular environment and by the studied cell type)[43,44]. Interestingly, when hydrogels were formed from gelatin and GH, but did not include the enzymatic crosslinker, minimal outgrowth was observed (Supplementary Fig. 12d, f). This suggests that the mere presence of integrin-based adhesion sites (on the gelatin) is not sufficient for migration and that some level of mechanical properties in the GR region is needed to generate traction for cells to move forward.

In addition to biophysical features in the hydrogel, biochemical signals are also important for migration. The polymers in our bicontinuous hydrogel have inherent adhesive sites that promote cell

**Fig. 4 | Migratory cells utilize adhesion-based mesenchymal migration strategies. a, b** Representative 5 µm maximum Z projection of a cell migrating through a 3 wt% GH hydrogel (left, GR domain: green, actin: magenta) and corresponding particle image velocimetry (right, colored arrows with magnitude normalized per frame) over time (**a**, Scale bar = 50 µm, labeled time denotes time elapsed from the beginning of imaging on day 3) and corresponding quantification of the average bead displacement vector contribution along the axis of migration within boxes in panel **a** over time (**b**, representative of 3 migrating cells across 2 biologically independent experiments, with cell corresponding to panel a denoted in red). **c** Schematic of cell migrating in 3 wt% GH hydrogel (numbered markers denote approximate phase of migration that correspond to panels **a**, **b**). **d** Representative cell (magenta) and corresponding 3D representation of top 25% of bead displacements, in which each arrow represents a discrete bead and color corresponds with magnitude (with arrows magnified 6x for visualization). Color bar = 40 µm. **e, f** Representative actin (magenta) images of MFCs at day 3 treated with soluble RGD (sRGD) compared to PBS control (**e**, Scale bar = 200 µm), and corresponding quantification (**f**). $n = 5$ (PBS) or 8 (sRGD) spheroids per condition from 2 biologically independent experiments. **$**p = 0.0013$; Two-tailed unpaired student's t-test. **g, h** Representative actin (magenta) images of MFCs at day 3 treated with 100 µM MMP inhibitor (MMPi) or DMSO control (**g**, Scale bar = 200 µm), and corresponding quantification (**h**). $n = 4$ (3%-DMSO), 6 (1%-DMSO), 9 (3%-MMPi), 10 (0,1%-MMPi), or 12 (0%-DMSO) spheroids from 2 biologically independent experiments. **$**p = 0.0024$; two-way ANOVA with Tukey post hoc. Data are mean ± s.d. Source data for (**b**, **f**, **h**) provided as a source data file.

attachment. Classically, cells adhere to gelatin via integrins and to hyaluronic acid via CD44 receptors[48]. To probe the role of these adhesion sites on migration, we introduced soluble RGD as a competitive inhibitor of integrin binding due to the presence of RGD moieties on gelatin that are exposed due to collagen denaturing[49]. This resulted in a significant reduction in cell outgrowth (Fig. 4e, f). This further supports that cells rely on adhesion-dependent mesenchymal-based cell migration in the bicontinuous hydrogels. Conversely, when spheroids were embedded in hydrogels consisting entirely of GH (HA only), minimal cell outgrowth was observed (Supplementary Fig. 12e, f), suggesting negligible or very weak CD44-hyaluronic acid interactions (although this finding may also be influenced by the biophysical properties of the hydrogel). This is consistent with previous observations in other covalently linked HA-based hydrogels[50,51]. Taken together, these results imply that cell migration through the bicontinuous hydrogel is complex and based on the range of biophysical and biochemical cues that are provided to cells during migration.

Since gelatin is protease-degradable and ECM degradation is one mode of 3D migration[52], we next explored the role of matrix degradation on migration in these bicontinuous materials. Specifically, we evaluated whether matrix metalloproteinase (MMP) activity is required for migration by applying a broad spectrum MMP inhibitor (Marimastat). To determine the appropriate inhibitor concentration, spheroids in 0% GH hydrogel (which relies entirely on MMP activity for cell outgrowth) were treated with varying concentrations of Marimastat for 3 days (Supplementary Fig. 13a), with 100 µM Marimastat resulting in significantly reduced cell outgrowth (Supplementary Fig. 13b). Applying this concentration to all hydrogels (0, 1 and 3% GH) resulted in a reduction in cell outgrowth in all cases. In particular, in the 3% GH bicontinuous hydrogels (which showed the greatest migration overall), 100 µM Marimastat resulted in a significant reduction in cell outgrowth. While reduced, it is notable that this MMP-inhibited amount of outgrowth remained substantially higher than that seen in the 0% formulation (Fig. 4g, h). This suggests that while degradation can increase the rate of migration in bicontinuous hydrogels, migration does not require MMP activity.

**The role of interfacial material presentation on cell migration**

The bicontinuous hydrogel structure introduces high interfacial surface areas that provide routes for cell migration. To better understand the impact of material composition of the interface on migration capacity, we utilized a modified sandwich culture to produce a fully defined interface to cells[53]. Here, spheroids were seeded atop a crosslinked gelatin substrate (utilizing our 0% GH hydrogel, crosslinked via 1 U/mL transglutaminase), and subsequently covered with another hydrogel, thereby creating a simple, planar 3D interface (Fig. 5a, Supplementary Fig. 14a). When no material was added, the cells from the spheroid quickly (within 1 day) egressed and spread along the surface of the gelatin as expected, given that the 2D presentation of an adhesive surface provides no barriers to cell migration (Fig. 5b–d)[54]. When a solution of gelatin and transglutaminase was added on top, however, enzymatic covalent crosslinks were formed

with the first layer during gelation, fully encapsulating the spheroid and abrogating outgrowth. In this scenario, spheroids are full encapsulated in 3D and protease degradation is needed for cell migration, similarly to our 0% GH hydrogel. Interestingly, the addition of a GH phase atop the gelatin substrate supported migration to the same extent as when no material was added, indicating that the physical network does not impede interfacial migration. To better probe the biochemical and biophysical features that are important for interfacial migration, we also added an inert, nonadhesive agarose layer on top. Under this condition, the extent of outgrowth was significantly higher than in our fully encapsulated group (Fig. 5c, d). Importantly, minimal outgrowth was observed perpendicular to the interface into the bottom gelatin layer, no matter the composition of the top layer (Supplementary Fig. 14a, b).

In our bicontinuous hydrogels, we noted a difference in mechanical properties between the GR and GP domains. Likewise, when the top layer of the sandwich model was agarose, we observed less migration compared to no material addition and GH addition, suggesting that the dynamic properties of the GH phase may be important for supporting migration[55]. To probe the role of differential mechanical properties of interfacial materials on cell migration, we placed agarose atop our spheroids with varied mechanics ranging from a modulus similar to the gelatin substrate (0.25% agarose) to a hydrogel that was 36 times the compressive modulus of the gelatin substrate (6% agarose, Supplementary Fig. 14c). Despite these differences in mechanical properties, we observed no differences in the degree of cell outgrowth occurring along the interface (Supplementary Fig. 14d, e). Again, minimal outgrowth perpendicular to the interface was observed Supplementary Fig. 14f). This suggests that it is the physical manifestation of the permissive interface itself, rather than the mechanics across the interface, that is most important for migration to occur. Overall, these results indicate that cells will rapidly migrate along interfaces between two hydrogel materials unless the hydrogels are crosslinked together to form a barrier to motility.

The unique combination of our two distinct subdomains led to a dynamic bicontinuous material with high 3D interfacial surface area. To test whether such rapid cell migration could be achieved in other bicontinuous hydrogel systems, we created fully defined 3D interfaces via granular hydrogel composites. In this case, interfaces could be tuned by varying the extent of jamming of particles within a continuous hydrogel that fills the void spaces. Specifically, we extrusion fragmented agarose particles, jammed them to various extents at increasing centrifugation speeds, and encapsulated them within a gelatin-only phase[56] (Fig. 5e, Supplementary Fig. 15 where 100 g, 500 g, and 9800 g correspond to low, medium, and high jamming/densification). When formed and characterized, the gelatin formed a continuous and connected domain across all structures (Fig. 5f, Supplementary Movie 7). Conversely, agarose particle connectivity (volume % within the greatest object in this domain) varied depending on the level of jamming – from ~50% for low speeds to ~100% for high speeds, and this corresponded to increases in interfacial surface area.

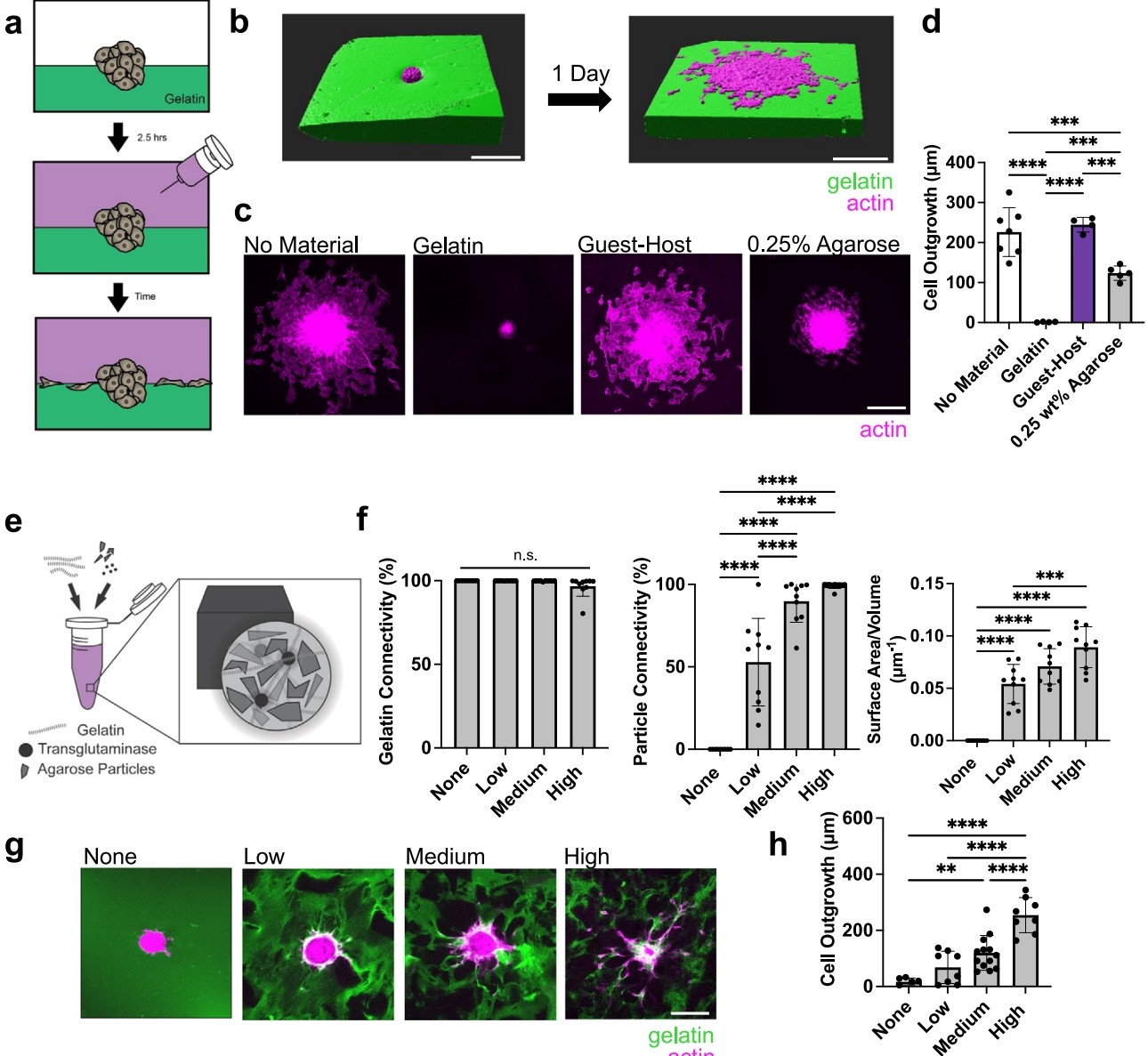

**Fig. 5 | The extent and composition of interfacial material presentation on cell outgrowth. a** Schematic of study design in which MFC spheroids are seeded on a gelatin substrate and additional material is placed atop to form an interface. **b** 3D reconstructions of the representative spheroid (magenta) from no material condition prior to (left panel) and after (right panel) spreading along the interface (gelatin: green). Scale bar = 200 μm. **c, d** Representative top view images of spheroids spreading along interfaces after 1 day (**c**, Scale bar = 200 μm) and corresponding quantification (**d**). n = 4 (GH), 5 (0.25 wt% Agarose), 6 (gelatin), or 7 (No Material) spheroids from 2-3 biologically independent experiments. No Material vs. Gelatin ****$p \leq 0.0001$; No-Material vs. 0.25 wt% Agarose ***$p = 0.0009$; Gelatin vs. Guest-Host ****$p \leq 0.0001$; Gelatin vs. 0.25 wt% Agarose ***$p = 0.0002$; Guest-Host vs. 0.25 wt% Agarose ***$p = 0.0006$; one-way ANOVA with Tukey post hoc. **e** Schematic of interface materials in which agarose particles undergo low, medium, or high densification within a continuous gelatin material. **f** Quantification of

connectivity of gelatin (left panel), connectivity of particles (middle panel), and interfacial surface area normalized to sample volume across hydrogels (right panel). n = 9 (None) or 10 (Low, Medium, High) regions across 3 hydrogels per condition. Left panel: n.s. indicates no statistical significance. Middle panel: None vs. Low, None vs. Medium, None vs. High, Low vs. Medium, Low vs. High, ****$p \leq 0.0001$. Right panel: None vs. Low, None vs. Medium, None vs. High, ****$p \leq 0.0001$; Low vs. High, ***$p = 0.0001$; one-way ANOVA with Tukey post hoc. **g, h** Representative Z-sections of MFC spheroids (actin: magenta) spread throughout interface materials (gelatin: green) at day 3 (**g**, Scale bar = 200 μm), and corresponding quantification (**h**). n = 6 (None), 8 (Low, High), or 13 (Medium) spheroids per condition from 2 biologically independent experiments. None vs. Medium **$p = 0.0042$; None vs. High, Low vs. High, Medium vs. High, ****$p \leq 0.0001$; one-way ANOVA with Tukey post hoc. Data are mean ± s.d. Source data for (**d, f, h**) provided as a source data file.

Strikingly, and similar to the bicontinuous gelatin and GH hydrogels, these granular agarose and gelatin hydrogels showed extensive cell outgrowth in formulations with high interfacial surface areas and minimal outgrowth when there was limited connectivity of one domain (Fig. 5g, h). These results validate the underlying premise that high interfacial surface areas and connectivity of two domains within a hydrogel are required to enable rapid cell migration.

## Bicontinuous structures promote cell migration from native tissue

While egress from MFC and MSC cellular spheroids offers some insight into how cells interact with the bicontinuous hydrogels, direct interactions with tissues provide additional insight into the utility of such materials for tissue repair. To assess this first ex vivo, living meniscus tissue explants were placed atop the bicontinuous hydrogels and

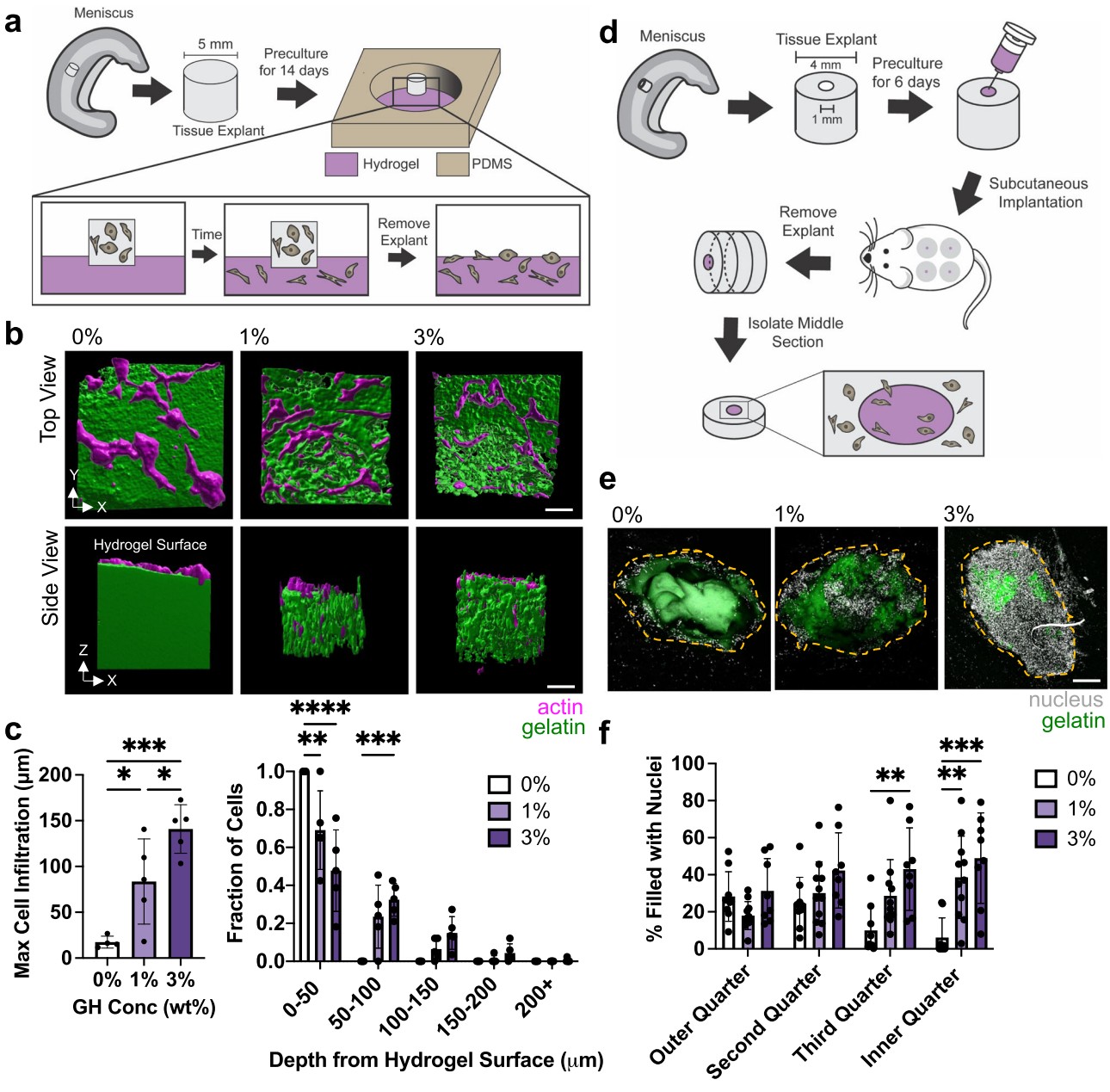

**Fig. 6 | Microinterfaces support increased cell infiltration ex vivo and in vivo.**
**a** Overview of ex vivo study design where meniscus explants are placed on top of each condition and cells are allowed to infiltrate the hydrogel. Modified with permission from[3]. **b** 3D reconstructions of actin (magenta)-GR domain (green) interactions after 6 days. Scale bar = 50 μm **c** Quantification of maximum infiltration (left, through determining 95% quantile) and extent of cell infiltration through hydrogel depth (right). $n = 4$ (0%) or 5 (1,3%) explants per condition from two biologically independent experiments. Left panel: 0% vs. 1% *$p = 0.0282$; 1% vs. 3% *$p = 0.0424$; 0% vs. 3% ***$p = 0.0004$; One-way ANOVA with Tukey post hoc. Right panel: 0–50: 0% vs. 1% **$p = 0.0021$; 0–50: 0% vs. 3% ****$p \leq 0.0001$; 50–100: 0% vs 3% ***$p = 0.0009$; Two-way ANOVA with Tukey post hoc. **d** Overview of in vivo study design where hydrogels are injected into meniscus explant defects and implanted subcutaneously into rats, with the middle region of the explant removed for analysis. **e** Representative images of nuclei (white) within hydrogels (green) after 14 days (orange dashed line denotes approximate defect edge). Scale bar = 200 μm. **f** Quantification of cell density within binned quartiles of defect space. $n = 8$ (0,3%) or 11 (1%) explants per condition across 8 rats. Third Quarter: 0% vs. 3% **$p = 0.0086$; Inner Quarter: 0% vs. 1% **$p = 0.0038$; Inner Quarter: 0% vs. 3%: ***$p = 0.0001$; two-way ANOVA with Tukey post hoc. Data are mean ± s.d. Source data for (**c**, **f**) provided as a source data file.

cultured for 6 days (Fig. 6a)[3]. For the 0% GH hydrogel, cells gradually migrated from the tissue explant onto the surface and spread, but did not penetrate into the hydrogel (Fig. 6b). Conversely, for the 1 and 3% GH hydrogels, cells egressed from the meniscal tissues and penetrated the hydrogel via microinterfaces, with greater extents of penetration with higher GH content (Fig. 6c). We found similar trends between 0% GH hydrogel and other uniform hydrogels, including collagen and acrylated hyaluronic acid (Supplementary Fig. 1), which all

demonstrated minimal cell infiltration when compared to the bicontinuous hydrogels (Supplementary Fig. 16). Consistent with the spheroid outgrowth studies, cells exiting native tissue were observed to migrate along the hydrogel microinterfaces (Supplementary Fig. 17). To extend these studies to the complex biochemical milieu of the in vivo environment, we next evaluated cell migration into hydrogels injected into defects within meniscal tissue that were subsequently implanted subcutaneously in athymic rats (Fig. 6d). After 2 weeks,

constructs were removed and the central regions were isolated and analyzed. While 0% GH hydrogels showed some cell invasion in the outer periphery of the hydrogel filling the defect, the innermost regions were largely devoid of cells (Fig. 6e, f, Supplementary Fig. 18). Conversely, the 1% and particularly the 3% GH hydrogels exhibited dense cellularity throughout the defect space. Further labeling suggested that infiltration from immune cells was limited, as only a small population of infiltrated cells exhibited CD68 staining, a macrophage marker (Supplementary Fig. 19). In monitoring the fluorescent signal from incorporated gelatin, there was similar fluorescence observed in all samples after 2 weeks (Supplementary Fig. 20), indicating that remodeling in vivo was similar across formulations with respect to gelatin degradation. These data support that bicontinuous hydrogels promote rapid migration from spheroids and tissue, in both controlled in vitro environments and within the complex in vivo environment, supporting their application for tissue repair scenarios.

## Discussion

Hydrogels have been used extensively to model cellular interactions with complex extracellular environments, and these studies have identified porosity, degradation, and viscoelasticity as biophysical cues necessary for 3D migration[6]. However, most of these previous studies employed monolithic 3D materials that are not representative of the complex microstructural heterogeneity seen within native tissue. Likewise, many studies that focus on interstitial 3D cell migration employ collagen gels, which begin as uniform materials and are remodeled into architectures that do not reproduce the complex features and microinterfaces in tissues that support migration[57]. For example, these platforms fail to recapitulate the interfaces found between tissues (e.g. collagen bundles) that support the rapid migration of cells such as metastatic cancer cells or matrix depositing fibroblasts[14–17]. Unfortunately, modeling this interfacial complexity in hydrogels has been limited by the challenges that arise with creating micron-scale interfaces from the bottom-up[58].

Bicontinuous hydrogels represent one potential solution. These materials rely on either arrested immiscibility or block copolymer assembly to introduce interpenetrating but distinct microdomains within a 3D structure[21,26]. While bicontinuous hydrogels have been fabricated for decades[59], and more recent studies have employed these unique structures to introduce fatigue resistance[27] or provide superior electrical conductivity[24], they are generally not compatible with cell-based studies due to the extreme temperatures[60] or organic solvents[61] required for their production. To fill this need, we combined modified biopolymers (i.e., gelatin and hyaluronic acid) to engineer a bicontinuous hydrogel through controlled immiscibility. The resulting domains were either gelatin-rich (GR) or gelatin-poor (GP), with structural features dependent on the initial solution concentrations and compositions, as well as the extent of mixing.

With this biopolymer-based bicontinuous hydrogel, we demonstrated that mixing and stabilization of immiscible phases could be conducted under cytocompatible conditions to engender distinct domains with high interfacial surface area and connectivity and bulk properties that varied with the introduction of the physical network (Fig. 1). Interestingly, with increasing bicontinuity, we noted a decrease in the Young's modulus with AFM-nanoindentation, but equivalent storage moduli with rheology. This discrepancy may be due to the differences in both the spatial scale and frequency of indentation at which the hydrogels were probed between these two mechanical tests (Fig. 2). The introduction of controllable microinterfaces supported rapid cell migration from spheroids and invasion by tissue-resident cells (Figs. 3 and 6), using the meniscus as an example. Importantly, we further defined the cell-material interactions occurring in the bicontinuous hydrogel, highlighting directional tracts as cells migrated, and showed how both adhesion and degradation regulate this process (Fig. 4).

Our bicontinuous hydrogels rely on controlled immiscibility to form their characteristic interconnected structures. These structures likely arise due to the strong physical interactions between our guest and host moieties that serve to arrest the interfaces temporarily, during which time the enzymatic crosslinking stabilizes these microdomains. To extend this strategy, other bicontinuous hydrogels could be fabricated using two immiscible constituents or employing phase separation (perhaps by engineering strong physical interactions or using inherently immiscible high molecular weight polymers) and utilizing alternate stabilization strategies (such as photoinduced crosslinking)[62]. Similarly, given the role of biopolymers in other block copolymer assemblies[63] for cell-material interaction studies, introducing bicontinuous hydrogels using biopolymer-based block copolymers may also provide a viable option. Experimentation with constituent ratios and immiscibility may be required, as simply adding two viscous mixtures (e.g., collagen, AD-HA, and CD-HA) was insufficient to obtain a bicontinuous structure (Supplementary Fig. 21).

Enhanced migration due to bicontinuity was replicated in alternate material systems (planar interfaces and granular composites) and established that the microinterfaces, rather than the differential mechanics of either phase or their specific chemical composition, controlled this migration process (Fig. 5). Increased migration in GH-based interfaces, when compared to agarose-based interfaces (Fig. 5), suggests that malleability of the interface constituents may alter the interfacial strength and ease of cell movement along microinterfaces, but further experimentation would be required to explore this hypothesis. Additionally, attempting to tune features such as scale and tortuosity of the interconnected domains, will require further refinements.

Taken together, our studies define a generalized governing principle of how bicontinuous material-mediated cell migration operates. These unique cell-compatible materials can be leveraged to expand the utility of current clinically used materials to develop materials that employ bicontinuity to define migratory paths to promote tissue regeneration and repair. Likewise, this concept may be further expanded to reprogram native tissue itself to influence endogenous cell migration, optimizing directed cell mobility in contexts where desirable (e.g., wound healing) and limiting it in scenarios where it is detrimental (e.g., cancer metastasis). Ultimately, this work expands our knowledge on the extracellular biophysical regulators of cell migration and defines material-guided microstructural formulations that can be used to exploit interfacial cellular behavior.

## Methods

### Material synthesis and characterization

**Macromer synthesis.** All reagents were obtained from Sigma-Aldrich and Fisher Scientific unless otherwise specified. Hydrogel macromers were prepared as described previously[64]. Briefly, for all polymers, sodium hyaluronate (HA, MW = 60 kDa, Lifecore Biomedical) was modified with tetrabutylammonium salt (HA-TBA). Ad-HA modification was performed through the reaction of 1-adamantane acetic acid (Ad) and (dimethylamino)pyridine (DMAP) and ditert-butyl decarbonate (Boc2O) for 20 h at 45 °C in anhydrous dimethyl sulfoxide (DMSO). CD-HA modification was performed through the reaction of Mono-(6-(1,6-hexamethylenediamine)-6-deoxy)-β-Cyclodextrin (Crysdot) and benzotriazol-1-yloxytris (dimethylamino) phosphonium hexafluorophosphate (BOP) for 3 h at RT in DMSO. AHA modification was performed through the reaction of acrylic anhydride at a pH 9–10 for 3 h at RT in DI water. All polymers were purified via dialysis, lyophilized and modification was confirmed using $^1$H NMR (Bruker Neo 400). AD-HA, CD-HA, and AHA of 12%, 20%, and 92% modification by $^1$H NMR, respectively, were used for all experiments (Supplementary Fig. 1, analysis performed in TopSpin and MestReNova).

**Bicontinuous hydrogel fabrication.** The GH concentration (0, 1, 3%) signifies the combined polymer weight percent, with a consistent ratio of 1:1 between adamantane and beta-cyclodextrin. Hydrogels were formed by separately preparing solutions of gelatin (0 or 5 wt%, gel strength 300, Type A from porcine skin) with desired AD-HA concentration and a separate solution that combined CD-HA with transglutaminase (enzymatic crosslinker). The two solutions were manually mixed at 25 °C and placed at 37 °C for at least 150 min to allow for full crosslinking prior to hydration. Pure GH hydrogels were similarly formed.

**Uniform hydrogel fabrication.** Stock solutions of 7.17 mg/mL fluorescent collagen solution was mixed by combining fluorescent collagen and 10 mg/mL collagen (Advanced BioMatrix) as previously[65]. Collagen was then diluted with PBS and neutralized with 1 NaOH and placed at 37 °C for at least 60 min for gelation prior to hydration. 5 wt% AHA hydrogels were fabricated by combining AHA with 10 mM MMP-degradable crosslinker (GCNSVPMSMRGGSNCG, Genscript) and 1 mM thiolated RGD (GCGYGRGDSPG).

**Collagen-GH hydrogel fabrication.** Collagen stock was mixed with AD-HA, and CD-HA was mixed with 1 N NaOH and PBS to obtain a neutral pH. Both solutions were mixed together and placed at 37 °C for at least 60 min for gelation prior to hydration.

**Structural characterization.** To assess structural heterogeneity, fluorescent-conjugated gelatin was added to bicontinuous hydrogels (0.5 wt%, from pig skin) and Z-stacks were acquired (Leica TCS SP5, 10x air objective – 1.44 μm/pixel x 5 μm/voxel, 25x water objective – 0.566 μm/pixel x 5 μm/voxel, or 63x oil objective – 0.115 μm/pixel x 1 μm/voxel). Fluorescent intensity profiles and quantification of coefficient of variation was performed using a custom MATLAB script. 3D rendering of bicontinuous hydrogel structure was conducted in Imaris, in which GR domains were fluorescently labeled, and GP domains were obtained from inversion of GR domain fluorescence. To label individual polymers, fluorescent peptides (GenScript) were covalently conjugated to methacrylated AD-HA and CD-HA as described previously[66]. Briefly, corresponding methacrylated polymer was dissolved in 0.2 M (pH ~8) triethanolamine buffer (TEOA) to a final polymer concentration of 2 wt%, with peptides achieving a final concentration of 5 mg of thiolated peptide/100 mg of methacrylated polymer. Reaction was allowed to occur overnight at 37 C. Polymers were then purified via dialysis, and lyophilized. Fluorescent polymers were doped into the hydrogel prior to gelation such that 0.4 wt% of the solution was fluorescently labeled.

3D object counter was performed on both GR and GP domains to obtain the volume, surface area, and number of discrete objects within a given 3D space. Volume fraction calculations were performed by dividing the total volume of either GR or GP domains by the theoretical total volume of the ROI. Connectivity was defined as the volume of the largest GR or GP discrete domain divided by the total volume of each respective domain x 100%. Total internal interfacial surface area was normalized by volume within each ROI. Normalized interfacial surface area was obtained by dividing by the volume of the ROI.

**Mechanical characterization (Rheology).** Hydrogels were formed as described and deposited on the bottom plate of a rheometer immediately after mixing. An AR2000 stress-controlled rheometer (TA Instruments) was fitted with a 20 mm diameter cone geometry and placed at a 26 μm gap. Hydrogels were allowed to crosslink by applying an oscillating torque at 1 Hz and 1% strain for 150 min at 37 °C unless otherwise specified. After polymerization, the stress relaxation of the gel was tested at 10% strain for 10 min and the creep-recovery of the gel was tested at 100 Pa for 10 min. The frequency dependence at 1% strain was also characterized from 0.01 to 100 Hz and strain sweeps at 1 Hz were characterized from 0.001% to 1000% strain.

**Mechanical characterization (atomic force microscopy – AFM-nanoindentation).** AFM-based nanoindentation (Bruker's Dimension Icon AFM) was utilized to quantify microscale mechanical properties of the formed hydrogels. Cantilevers (HQ:CSC38/Cr-Au MikroMasch) had a nominal spring constant of 0.03 N/m and a 10 μm diameter spherical bead was attached to the tip extremity. Bicontinuous hydrogels were fabricated in between two glass coverslips with ~100 μm spacing. To query the mechanical properties of the hydrogel interior, the bicontinuous hydrogel was formed in a 1 mL blunt syringe and allowed to undergo complete gelation. The hydrogel was cut in half with a razor blade and AFM-nanoindentation was conducted on the newly exposed surface. Bicontinuous hydrogels were probed with an approaching and retraction rate of 10 μm/s in PBS, following established procedures[67]. Deflection error versus height sensor curves were collected in an 8 × 8 grid of 105 μm by 105 μm. The spring constant of the cantilevers were calibrated through thermal resonance frequency prior to the experiment. With an assumption of Poisson's ratio of 0.49 for highly swollen hydrogels, effective indentation modulus, $E_{ind}$, was obtained by fitting a Hertz model to the entire loading portion of each indentation force-depth curve[68]. Force map visualization and quantification of coefficient of variation was conducted with a custom MATLAB (MathWorks) script.

To probe mechanical properties within distinct GR and GP domains, 50 μm fluorescent beads (CD Bioparticles) were embedded at a 0.5 wt% concentration as fiducial markers. 3 × 8 grids of 30 μm by 105 μm were obtained. After indentation, the same hydrogels were imaged using confocal microscopy. The location of fiducial markers (beads) that appeared on the bright field AFM microscope were correlated to the location of beads within the confocal images to determine which fluorescent regions were probed (Supplementary Fig. 7b, c). Regions were manually denoted as GP or GR (or inconclusive if they were at or close to an interface; these regions were excluded from analysis). Outliers in AFM force maps were removed with the IQR method.

**Mechanical characterization (dynamic mechanical analysis).** Hydrogel prepolymer solution (30 uL) was placed into a cylindrical mold (4.78 mm diameter) and allowed to gel (gelatin at 37 °C for greater than 150 min, agarose at 4 °C for 1 h and, then transferred to 37 °C for 1 h). Uniaxial compression testing was conducted (TA Instruments Q800 DMA), with hydrogels exposed to a preload force of 0.001 N and compressed at a rate of 0.05 N min⁻¹. Compressive moduli were calculated as slopes between 10 and 20% strain on the stress-strain curve.

**Degradation studies.** 3% bicontinuous hydrogels were fabricated and added to PDMS molds (6 mm diameter, 6 mm height). Every two days, hydrogel volumes were measured and supernatant was acquired for uronic acid release assays. Uronic acid content was characterized via the uronic acid carbazole reaction according to the previous literature[69]. Briefly, a standard curve was prepared using sodium hyaluronate dissolved in water at 1 mg mL⁻¹ and subsequently serially diluted to produce a 7-point curve (1 mg mL⁻¹ to 0.03125 mg mL⁻¹, plus 0 mg mL⁻¹). To initiate the reaction, the sample (50 μL) was heated to 100 °C for 10 min immediately after the addition of 25 mM sodium tetraborate (200 μL, dissolved in sulfuric acid) and allowed to cool to room temperature for 15 mins. Next, 0.125% (w/v%) carbazole (50 μL, dissolved in absolute ethanol) was added to each well, heated to 100 °C for 10 min, and cooled to room temperature for 15 min. Sample absorbance was read immediately at 550 nm on a Tecan M200 Pro plate reader. A linear regression curve fit was used to determine uronic acid concentration. Data is presented as cumulative uronic acid release %.

## In vitro experiments

**Spheroid encapsulation in 3D.** Meniscal fibrochondrocytes (MFCs) were harvested from the medial meniscus of dissected juvenile bovine joints (Research 87) as described previously[70]. Briefly, menisci were minced into ~1 mm³ cubes and placed in tissue culture plates, where MFCs gradually emerged from tissue over two weeks. Mesenchymal stromal cells (MSCs) were isolated from juvenile bovine bone marrow[71]. All cells were cultured until 80% confluency of initial colonies and then stored in liquid nitrogen (90% FBS, 10% dimethyl-sulfoxide). For in vitro studies, MFCs and MSCs were used between passage 2-6. Spheroids (1000 cells/spheroid) were formed through 48-h culture on agarose (standard gelling temperature) microwell pyramid arrays (molds formed from AggreWell™ 400, Stem Cell Technologies). Spheroids were removed from microwells via pipette aspiration and mixing, allowed to settle, and then were transferred to hydrogels for assessment of outgrowth over time. Spheroids were added at a concentration of 1000 spheroids/mL of hydrogel solution, and hydrogels were completely gelled (for 150 min with transglutaminase) prior to adding media. For studies that consisted of networks with 5 wt% gelatin, 1 U/mL of transglutaminase and 3% soluble HA, an additional thin layer of 0% GH hydrogel was added atop these hydrogels to form a barrier to soluble HA dissolution from hydrogel. Unless otherwise noted, cells and spheroids were cultured in high glucose DMEM, supplemented with 10% Fetal Bovine Serum (FBS) and 1% penicillin and streptomycin at 37 °C and 5% $CO_2$.

**In vitro imaging and analysis.** Unless otherwise specified, Z-stack images were acquired on an upright confocal microscope (Leica TCS SP5).

All viability studies were conducted by treating cells with calcein AM (2 µM) and ethidium homodimer-1 (4 µM) for 3 h at 4 °C. For actin staining, all constructs were fixed in 10% neutral buffered formalin for 30 min at RT, permeabilized with 0.1% Triton X for 30 min at RT, and then stained at 1:1000 with phalloidin-647 to visualize the actin cytoskeleton in 1% bovine serum albumin (BSA) overnight at 4 °C. Actin staining was performed after 1, 2, and 3 days of culture. Analysis was carried out in ImageJ (NIH), where outgrowth was determined by manually denoting original spheroid body and extrapolating spheroid radius, with live stain used for spheroids in guest-host only or gelatin and guest-host hydrogels, and actin stain used for all other studies (Supplementary Figs. 10, 11b). Eight lines were drawn from the center of spheroid towards the periphery of the outgrowth. These values were then averaged and spheroid radius was subtracted to yield average cell outgrowth from a single spheroid.

For MMP inhibition studies, spheroids were embedded in 0% GH hydrogels (5 wt% gelatin) and allowed to migrate for 3 days to determine a concentration of Marimastat that was sufficient to reduce cell outgrowth. Constructs were treated with 0, 1, 10, 100, and 1000 µM Marimastat. After identifying 100 µM as the desired concentration, all experimental groups were treated with Marimastat and a control of DMSO for 3 days and stained for actin. Outgrowth was analyzed as above. For RGD binding perturbation, soluble RGD (5 mM RGD peptide, GCGYGRGDSPG, Genscript) was added to media in the 3% experimental group and outgrowth after 3 days was compared to PBS controls, with outgrowth analyzed as above.

For proliferation, the constructs were fixed and soaked in sucrose overnight and then embedded in OCT and flash frozen in liquid nitrogen-cooled 2-methylbutane. Using a cryostat microtome, constructs were sectioned to 30 µm thickness. Slides were then rehydrated for 5 min with Phosphate Buffer Saline (PBS), permeabilized with 0.1% Triton X for 5 min and then blocked with 3% BSA for 1 h at RT. Ki67 antibodies (2 ug/mL, Abcam ab15580) were added in 3% BSA to the sections and incubated at 4 °C overnight. Sections were rinsed with PBS for 5 min three times. The secondary antibody Alexa Fluor 647 IgG H&L (1:1000, Invitrogen A32733TR) was added in 3% BSA for 1 h. Slides were then rinsed three times again for 5 min each and mounted with Prolong Gold Antifade reagent with DAPI overnight at RT prior to imaging. In ImageJ, the number of Ki67+ cells was divided by the total number of cells (Moments threshold) to obtain the fraction of pro-liferating cells that had migrated from the spheroids. Antibody vali-dation was performed by seeding MFCs onto glass substrates for 3 days and imaging with an upright epifluorescent microscope (Sup-plementary Fig. 22).

To assess nascent protein deposition, studies were performed as previously described[51,72]. Briefly, the media consists of glutamine, methionine, and cysteine-free high glucose DMEM, supplemented with 0.201 mM cysteine, 50 ng/mL of Vitamin C, 100 µM L-methionine, 100 µg/mL sodium pyruvate, 8 mM Glutamax, 1% P/S, 10% FBS and 0.1 mM azidohomoalanine (AHA). At the desired time points, hydro-gels were washed, and stained for 30 min with CellTrackerRed in 3% BSA at 5 µg/mL at 37 °C and 5% $CO_2$. The gels were then washed several times with 3% BSA and stained with 30 µM DBCO-488 in 3% BSA for 30 min at 4 °C and washed 3 times again prior to fixation in 10% for-malin for 30 min at RT. Constructs were imaged as described previously.

**Spheroid outgrowth along 2D interfaces.** Gelatin hydrogels (5 wt%, 1 U/mL transglutaminase) were prepared in 8 mm biopsy-punched PDMS wells and allowed to fully gel. MFC spheroids were placed atop gelatin hydrogels and allowed to adhere for 2 h. Afterwards, gelatin (5 wt%, 1 U/mL transglutaminase), GH (3 wt%) or molten agarose (0.25%, 1, 3, and 6 wt%, low gelling temperature) were placed atop gels. After gelation, media was added, and constructs were fixed, stained for actin, and imaged after 1 day. Outgrowth parallel to the 2D interface was analyzed as above. Outgrowth perpendicular to 2D interface was quantified in Imaris. Briefly, cells (spheroid body and outgrown cells) were united into a single surface and fit into an object-oriented bounding box. The box dimension was obtained to determine the extent of perpendicular outgrowth, where the perpendicular out-growth diameter was matched to the shortest principal axis. As a result, outgrowth perpendicular to interface data includes spheroid radii.

**Live cell outgrowth.** Live-imaging studies were performed with a laser scanning confocal microscope (Nikon Ax 1) with a live-cell chamber (37 °C, 5% $CO_2$). For outgrowth studies, a 20X water immersion objective (0.38 µm/pixel, 5 um/voxel) was used. Formed spheroids were stained with deep red Cell Mask Plasma membrane stain (5 µg/mL) and embedded along with 0.2 µm-diameter fluorescently labeled beads at 0.7% (v/v) in the Ad-HA/gelatin prepolymer solution and 0.3% (v/v) in the CD-HA/transglutaminase prepolymer solution. After spheroid encapsu-lation and gelation, gels were incubated in live imaging media (Fluoro-Brite DMEM, 10% FBS, 1% P/S). Z-stacks (100 µm) were acquired every hour for 3 days. Imaging parameters were adjusted to minimize photo-bleaching and maximize cell viability.

Confocal time-lapse z-stacks were analyzed with Imaris 10.0. Migration track quantification was obtained through spots analysis of cells. Cells were estimated to be 25 µm spots and tracked via auto-regression motion algorithms, in which the maximum path length was set to 100 µm with a 5-spot gap max to predict future spot positions. Cell tracks were filtered and removed ( <30 µm) to reduce artifact motion. Tracks that appeared from outside field of view were removed manually to ensure that only cells that migrated from spheroid were evaluated. Migration tracks were rendered with a custom MATLAB script, with each cell track beginning from a common origin point. For real-time outgrowth renderings, the spheroid bodies were rendered as surfaces and cells were rendered as surfaces and overlayed with quantified tracks in Imaris.

**Anisotropic persistent random walk model.** The statistical char-acteristics of cell migration in our hydrogel system were studied using

the anisotropic persistent random walk (APRW) model[37,73]. According to this model, we first projected our 3D cell trajectories and obtained their corresponding $x$ and $y$ coordinates. Then, we verified that the cell motility is time-invariant after 30 h in our system (Supplementary Fig. 9a) and thus APRW model can be applied. We next calculated the MSD from the cell trajectories at different time lags as follows:

$$\text{MSD}(\tau) = <[x(t+\tau) - x(t)]^2 + [y(t+\tau) - y(t)]^2> \qquad (1)$$

where $\tau$ represents the time lag and $<\cdots>$ shows time averaging. The angular displacements $d\theta$ between two consequent steps of movement with time lag $\tau$ were then calculated using the definition of inner product of the two velocity vectors as follows:

$$d\theta(t,\tau) = \arccos\left(\frac{v_t \cdot v_{t+\tau}}{|v_t||v_{t+\tau}|}\right) \qquad (2)$$

where $|\ldots|$ shows the vector magnitude. After that, we obtained the primary and non-primary axes of migration for each individual cell through singular value decomposition (SVD) analysis of its velocity matrix. We subsequently calculated average magnitudes of the cell velocities at different orientations relative to the primary axis and obtained the corresponding velocity magnitude profiles. According to the APRW model, cell motility is assumed to display different persistence random walks along the two axes of migration. As a result, we calculated the MSDs of cell movements along each of these orthogonal migration axes and then fitted them to the following equations:

$$\text{MSD}_p(\tau) = S_p^2 P_p^2 \left(e^{-\frac{\tau}{P_p}} + \frac{\tau}{P_p} - 1\right) + 2\sigma^2 \qquad (3)$$

$$\text{MSD}_{np}(\tau) = S_{np}^2 P_{np}^2 \left(e^{-\frac{\tau}{P_{np}}} + \frac{\tau}{P_{np}} - 1\right) + 2\sigma^2 \qquad (4)$$

where $\sigma^2$ is the variance of observation noise in cell position and $S_p(S_{np})$ and $P_p(P_{np})$ show the cell speed and its persistent time along $\vec{\mathbf{p}}$ ($\mathbf{n\vec{p}}$) direction. The obtained values for the cell speeds and persistent times can be further used to calculate cell diffusivities in different migration directions using the following formula:

$$D_p = \frac{S_p^2 P_p}{4} \qquad (5)$$

$$D_{np} = \frac{S_{np}^2 P_{np}}{4} \qquad (6)$$

The ratio of these cell diffusivities gives the anisotropy index of the cell migration:

$$\phi = \frac{D_p}{D_{np}} \qquad (7)$$

and the total cell diffusivity can be calculated as $D_t = D_p + D_{np}$. Finally, based on the obtained cell speeds and persistent times, cell trajectories were also simulated via the governing stochastic differential equations of the APRW model and the corresponding MSDs and velocity magnitude profiles were extracted to verify the model accuracy. The details of the governing differential equations and the simulation process are described further in Wu et al.[37].

**Bead displacement studies.** Cells were allowed to migrate from spheroids and were imaged at 10 min intervals for 3 h on day 3 with a 40x water immersion objective (0.22 μm/pixel, 1 μm/voxel). Cells that

migrated along one axis throughout the imaging time span were selected for further analysis. 3D stacks were drift corrected with ImageJ Plugin Correct 3D Drift. A stack of 5 μm was isolated based on a cell of interest, max projected, and oriented such that cells were migrating in the negative direction of the axis of migration. HyperStackReg was used as a second drift correction and bead displacements were quantified with the ImageJ PIV plugin (1st interrogation window size: 128 pixels, 1st search window size: 256 pixels; 2nd interrogation window size: 64 pixels, 2nd search window size: 128 pixels). A region of interest (40 μm x 150 μm - which corresponds to 2x the average cell volume) was manually identified in the forward path of cell migration. Vertical components of bead displacement vectors in this region of interest were averaged and then plotted based on time from the beginning of imaging session on day 3.

**Visualization of 3D bead displacement.** To remove noise, bead images were background subtracted in ImageJ with a 4-pixel radius. To extract the bead positions, we used a maximum likelihood estimator-based Lucy-Richardson deconvolution method. Bead displacements were then computed using a topology-based particle tracking (TPT) algorithm implemented in MATLAB[74]. To extract the cell body for 3D reconstruction, we first background subtracted the images in ImageJ with a 50-pixel radius and segmented the images using the Otsu's method. Then, 3D objects were detected using the bwmorph3 function in MATLAB, followed by a size-exclusion procedure to remove cell debris. The displacement field was enlarged 6 times and overlayed with the cell body for visualization purposes.

**Gelatin-agarose particle composite hydrogels.** Extrusion fragmented agarose particles were prepared as described previously[56]. Briefly, 3 wt% agarose was cooled at 4 °C for 1 h, and then sequentially extruded through 18, 21, 23, 25, 27, 30, and 34 G needles. Particles were resuspended in gelatin solution (gelatin - 5 wt%, transglutaminase - 1 U/mL) and then centrifuged at 100, 500, and 9800 rcf (low, medium, and high density) for 30 seconds. Supernatant was aspirated and gelatin-microgel slurry was combined with a spheroid pellet, manually mixed, and then added to 4.78 mm molds. The construct was allowed to gel for 150 min prior to addition of media. After 3 days, constructs were fixed and analyzed for cell outgrowth as above. Structural analysis of composite hydrogels was conducted as above (where agarose particles were obtained by inverting gelatin fluorescence).

**Ex vivo and in vivo experiments**

**Ex vivo studies.** Using biopsy punches, cylindrical tissue explants (5 mm in diameter and 1 mm in height) were excised from the middle zone of the meniscus such that circumferential fibers were oriented normal to the surface of the hydrogel. Explants were subsequently incubated in basal media for 2 weeks to support cell occupation of the explant periphery[3]. Afterwards, hydrogels were fabricated within 8 mm diameter PDMS molds and explants were placed on top of hydrogels and cells were allowed to egress from the tissue onto gels for 6 days, at which point explants were removed, and hydrogels (and their infiltrated cells) were fixed as above. Constructs were then stained with Hoechst 33342 (20 μM) to identify nuclei and actin (as above). Cell infiltration quantification and 3D construct rendering were carried out in Imaris.

**In vivo studies.** To evaluate the effect of scaffolds on meniscal repair in an in vivo setting, a nude rat xenotransplant model was employed. All procedures were approved by the Animal Care and Use Committee of the University of Pennsylvania Office of Animal Welfare (Protocol 806580). Adult bovine meniscal explant cylinders (4 mm outer diameter, 1 mm inner diameter, $n = 10$ donors) were biopsy punched as previously. The defect was filled with 0, 1, or 3% GH hydrogel via syringe injection. Explants were allowed to incubate for at least

150 min before subcutaneous implantation into male nude rats (N = 8; NTac: NIH-*Foxn1*$^{rnu}$, 9-12 weeks old, ~300 g, Taconic). Rats were anesthetized with isoflurane and the dorsal area shaved and scrubbed with chlorhexidine and betadine. Four subcutaneous pockets were made with 1 on the cranial and caudal aspects on each side of the midline via blunt dissection. One construct per group was inserted into each pocket (with a repeat of one of the groups in each rat). The incision was closed with wound clips. At 2 weeks, rats were euthanized by $CO_2$ asphyxiation and the constructs were removed from the subcutaneous tissue. Constructs were cut into 1 mm sections using a custom meniscus slicer and the slices at the end of the constructs were removed to minimize contributions from host cell infiltration. Constructs were fixed for 15 min at 37 °C and stained with Hoechst for 30 min. To assess cellularity, cell signal was Otsu thresholded in ImageJ and % positive cell area was quantified within each quartile of the manually segmented total defect area. To assess fluorescence loss, fluorescence % area was thresholded and subtracted to obtain % defect without fluorescence. Any constructs in which gel in defect did not remain intact (through lack of fluorescence and cells) were excluded from analysis. To assess CD68 labeling, constructs were flash-frozen and cryosectioned to 10 μm thickness as previously, and stained with CD68 (1:500, MCA341GA clone ED1; Bio-Rad) and secondary antibody Alexa Fluor 546 IgG H&L (1:500, Invitrogen) prior to mounting with DAPI. Imaging of CD68 was conducted with a Zeiss Axioscan Z1 slide scanner using a 40X objective and a Colibri 7 LED illumination source. In ImageJ, the number of cells positive for CD68 were divided by the total number of cells to obtain the fraction of CD68+ infiltrating cells.

**Statistics and data presentation.** All statistical analyzes were performed using Microsoft Excel and GraphPad. Unless otherwise specified, the robust regression and outlier removal method was used to remove outliers prior to performing statistical tests. Significance was assessed with two-way student's t-test (for 2 experimental groups) or by one or two-way ANOVA with Tukey's HSD post hoc testing, where a *p* value < 0.05 was considered significant. When no experimental groups were significant, n.s. was used to denote no statistical significance. In ANOVAs, when some groups were significant, a straight line between significant groups was denoted, and unlabeled groups in these graphs have no statistical significance. N values and biological replicates are described in figure captions. All data are reported as mean ± standard deviation as denoted in the figure caption. All statistical test p, q, t, and DF values are reported in Source Data. Schematics were designed with Adobe Illustrator.

### Reporting summary
Further information on research design is available in the Nature Portfolio Reporting Summary linked to this article.

## Data availability
All the data generated or analyzed during this study are included within this article, its Supplementary Information and its Source Data. Additional information is available from the corresponding author on request. Source data are provided with this paper.

## Code availability
The custom code for the APRW model is available at https://github.com/MohammadDehghany/Nature-Communication-APRW-Model [https://doi.org/10.5281/zenodo.10766171][75]. The custom code for analyzing 2D bead displacement and in vivo cell density is available at https://github.com/klxu1/Xu_NatureComms24_Bicontinuous. [https://doi.org/10.5281/zenodo.10775211][76]. Codes for 3D bead displacement visualization are available upon request due to specific tailoring towards individual application.

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

## Acknowledgements

The research was funded by the National Institutes of Health (grants no. R01AR056624 and R01AR077362 to JAB, RLM, R01CA221346 to MW, F30AG074508 to KLX), the Penn Center for Musculoskeletal Disorders (grant no. P30 AR069619), the National Science Foundation (grant no. CMMI-1548571 (Science and Technology Center for Engineering Mechanobiology) to JAB, RLM, CMMI-1751898 to LH). The authors thank the Penn NMR facility, the Penn CDB Microscopy Core Facility, the Penn Center for Musculoskeletal Disorders Histology Core and the CU Boulder BioFrontiers Advanced Light Microscopy Core for use of their instrumentation and technical support. The authors would additionally like to thank Dr. Claudia Loebel, Dr. Elizabeth Kahle, Dr. Xi Jiang, Dr. Ryan Locke, Dr. William Querido, Dr. Nancy Pleshko, Riti Sharma, Dr. Ottman Tertuliano, Dr. Rebecca Wells, and Dr. Paul Janmey for helpful discussions.

## Author contributions

K.L.X., R.L.M. and J.A.B. designed the project. K.L.X. performed experiments and conducted data analysis with assistance from N.D., L.L. and Y.Z. The data regarding the structural properties of hydrogels was analyzed and interpreted data by N.D., including Imaris analysis of real-time cell migration. H.F. and L.H. analyzed and assisted with the interpretation of AFM studies. M.D. and V.S. conducted computational analysis of cell migration and interpreted findings. M.D.D. assisted with acquiring live imaging studies and providing experimental input. B.C. and M.W. assisted with designing and interpreting bead displacement studies. K.L.X., R.L.M. and J.A.B. wrote the manuscript. All authors discussed results, assisted in data interpretation, and provided input on the paper.

## Competing interests

The authors declare no competing interests.
