## [Peer Review File · Nature Communications]

REVIEWER COMMENTS

Reviewer #1 (Remarks to the Author):

The manuscript by Xu et al. reported a microinterface-facilitated cell migration in bicontinuous hydrogel. The experiments are well-designed and performed. The following points should be considered and addressed before it is suitable for publication.

1. It is abnormal that G'' of 0% sample does not increase in Fig. 1c during gelation.
2. It is necessary to show and compare the viscoelastic properties of pure gelatin gel and GH physical gel, corresponding to the GR and GP phases in the bicontinuous gel, respectively.
3. A very wide range of domain sizes in some pictures (see Fig. 3b) can be observed. Does the mixing intensity affect the bicontinuous morphology?
4. The modulus in Fig. 2e can show some difference between the modulus at different points. However, it is a clear way to show the modulus difference in GR and GP phases. A modulus map superposed on the morphology can make the difference more clear.
5. A control experiment using the pure GH physical gel for cell migration is necessary to prove that the rapid migration in the bicontinuous gel is not due to the low modulus of the GP phase.
6. Although it is stated that cells use the interfaces between the GR and GP domains to migrate, direct observation in Fig. 3 is somewhat difficult. Moreover, such a statement implies that the interfacial area between two domains is readily broken by the cell, i.e., the interfacial strength is low, which can be evaluated using the bilayer structure in Fig. 5. The difference in the outgrowth in Fig. 5c using guest-host gel and agarose gel indicates the possible difference in the interfacial strength.
7. In the bilayer experiments (Fig. 5), it is necessary to compare the cell outgrowth in the direction perpendicular to the interface. The conclusion is solid only when vertical outgrowth is much smaller than parallel.

Reviewer #2 (Remarks to the Author):

The work by Xu et al. presents a design of non-miscible biopolymer to obtain discrete, yet continuous microstructural features in one mixed hydrogel. This biopolymer is applied toward the study of cell migration in the context of possible regenerative medicine application. The resulting microinterface

promote cell migration along its boundary. They proceed to demonstrate that the material has pro-migratory properties in vivo. This is certainly a highly relevant way to achieve cell migration with contact guidance using material formulation. There are, however, a number of issues that need to be addressed.

Main concerns:

1- There is currently no real discussion apart from a few experimental justifications provided in the results section. The authors need to better situate their findings and the relevance of this material in light of the advances in the field and to provide context to non-experts. The provided outlook is certainly not sufficient.

2- The stiffness measurement maps in Fig 2 are difficult to interpret.

3- I am quite confused by the diverging results for the mechanical test. On one hand, the storage modulus remains the same for all formulation while the loss modulus increased with %. On the other hand, average AFM measurements indicate that the modulus decrease with %. I would expect the two measurement methods to be somewhat correlated. Is there something I am missing here?

4 – It appears that the AFM measurements were performed on the upper surface of these hydrogels. Is it possible that this hydrogel/ air or liquid interface could exhibit altered boundary condition / mechanical properties? It should be possible to process this kind of hydrogel for cryosection (maybe using two color OCT) and then perform AFM on the internal structure of the hydrogel.

5- The claim that bicontinuous hydrogels may provide energetically favorable paths for cell migration is a bit bold given that there is no direct evidence provided for this.

6- Given the impact of the MMP inhibitor, do the authors know what is the actual porosity of the GH side?

7- The use of the RGD peptide here is surprising. Wouldn't it be better to use the GOFER motif instead since gelatin and not a fibrinogen is used in the system?

8- The explant and in vivo experiments are quite interesting and relevant and show that cells will readily invade within the hydrogel. However, it is unclear which kind of cells actually enter the hydrogel, particularly in the in vivo experiment. Are these chondrocytes, immune cells or something else?

9- For those two experiments, is the bicontinuous hydrogel more efficient at driving cell migration compared to other uniform hydrogels (collagen for instance)? Performing such an experiment would provide a good comparison point for the bicontinuous material.

10- For the in vivo conditions, how much of the material is still intact at the end of the experiment? Are the invading cells replacing some of the material with native ECM?

Minor concern:

1- Appreciating how the micro-interface support cell migration is difficult with the current pictures in the different figures. Showing higher magnification pictures (e.g. insert) of the migrating cells would be helpful here.

2- Is it possible to obtain a similar bicontinuous effect if gelatin is replaced with full-length collagen?

Reviewer #3 (Remarks to the Author):

In this manuscript, Xu and co-authors describe the fabrication of a hydrogel by mixing hyaluronic acid (HA)-cyclodextrin (CD) and hyaluronic acid (HA)-adamantane (AD) with highly crosslinked gelatin. The authors claim the formation of a bicontinuous hydrogel and demonstrate cell migration behavior into this unique structure. The manuscript is well-written and represents a thoughtful experimental design. This manuscript is a strong candidate for publication in Nature Communications once the concerns raised by the reviewers are addressed by the authors.

1. It appears that the bicontinuous feature, which arises from the combination of HA-AD, HA-CD, and gelatin, is a defining characteristic of the material. Therefore, it would be appropriate to include this information in the title and the abstract. Mentioning only gelatin and HA (instead of HA-AD and HA-CD and crosslinked gelatin) in the abstract may be somewhat misleading.
2. The authors mixed transglutaminase and hyaluronic acid (HA)-cyclodextrin (CD) with hyaluronic acid (HA)-adamantane (AD) and gelatin, claiming the formation of a bicontinuous hydrogel due to limited miscibility of AD and CD. While it's clear that interactions between HA-AD and HA-CD resulted in phase separation (as demonstrated in extended Fig. 2c), it's essential to determine if there are any additional phases beyond what is observed through confocal imaging. What evidence supports the conclusion that HA-CD and HA-AD together form only one continuous phase?
3. The authors used a relatively high transglutaminase concentration (1U/mL) with 5 wt% gelatin for cross-linking. This may result in a densely crosslinked gelatin network, making it challenging for cells to migrate. It seems that using a lower transglutaminase concentration might yield a different outcome, potentially allowing cells to migrate through the gelatin-rich region as well. Did the authors experiment with different transglutaminase concentrations to create the bicontinuous hydrogel?
4. Fig. 6e shows extensive material degradation for 3%, along with a notably high cellularity (also evident from supplemental Fig. 7). These cells could potentially be different types of immune cells infiltrating the site of material degradation, triggering inflammatory reactions. Have the authors identified the type of cells that infiltrated the defect site? Simple histological analysis could provide insight into the inflammatory environment. Given this situation, the bicontinuous hydrogel may not be suitable for in vivo applications.

5. The bicontinuous hydrogel appears to have low mechanical strength. The authors did not provide any degradation test results (both in vitro and in vivo). The evidence from the in vivo results presented in Fig. 7 suggests that this hydrogel may be highly unstable. It would be valuable for the authors to discuss the potential applications of this hydrogel in tissue engineering considering its mechanical properties and instability.

6. Do the authors anticipate any potential limitations in the fabrication of a bicontinuous hydrogel using alternative materials? What challenges might other researchers encounter in their efforts to develop a bicontinuous hydrogel?

Dear Editor and Reviewers,

We are grateful for the positive and thorough review of our work. Based on this feedback we have carried out additional experiments and have extensively modified the original manuscript. Point-by-point responses to each reviewer are provided below and changes are highlighted in the revised manuscript. We believe that this feedback has considerably strengthened the work towards making a significant impact on the field of engineered biomaterials to support rapid cell migration.

Reviewer #1

The manuscript by Xu et al. reported a microinterface-facilitated cell migration in bicontinuous hydrogel. The experiments are well-designed and performed. The following points should be considered and addressed before it is suitable for publication.

1. It is abnormal that G'' of 0% sample does not increase in Fig. 1c during gelation.

Response: We thank the reviewer for raising this point for clarification. Our 0% formulation consists of only gelatin that is crosslinked with transglutaminase. The system is highly viscoelastic at short times based on the physical association of gelatin and then there is an increase in G' over time as transglutaminase introduces additional covalent crosslinks into the gelatin hydrogel. There is a small increase in the loss modulus during this time (see Figure R1 below for 3 examples of this gelation process) where there are slight increases in G'' observed; however, this is much lower than the formulation that incorporates the guest host chemistry.

To help make this point clearer, we have added the following text “A slight increase in loss modulus was observed over time across increasing GH concentrations; but G'' in the 0% GH hydrogel remained much lower than that observed when GH was incorporated, likely due to the dominance of covalent crosslinking.” and have replaced **Fig. 1c** with a more representative time sweep for the 0% hydrogel that illustrates this modest increase in G'' .

Figure R1: Representative examples of 0% hydrogel gelation kinetics.

2. It is necessary to show and compare the viscoelastic properties of pure gelatin gel and GH physical gel, corresponding to the GR and GP phases in the bicontinuous gel, respectively.

Response: We agree with the reviewer that understanding the individual components of our bicontinuous hydrogel is important. We have conducted extensive work on rheological characterization of the pure gelatin and bicontinuous hydrogels (as shown in **Fig. 1c,e** and **sFig. 4**), which now includes a comparison between the 0% GH hydrogel and a 3 wt% GH hydrogel (**sFig. 4g,h**). These data show the G' , G'' , and $\tan(\delta)$ for the 0% hydrogel (pure gelatin with

transglutaminase enzyme crosslinker) and GH hydrogel, as well as a frequency sweep (1% strain) of the GH hydrogel and highlights that the GH has increased viscoelasticity (that is, increased $\tan(\delta)$ and mechanical properties that vary with frequency) compared to the enzymatically crosslinked gelatin hydrogel.

To clarify this, the following additional text has been added to the manuscript regarding these data: “Additionally, comparison of the 0% GH hydrogel (gelatin-only) and GH-only hydrogel highlights the increased viscoelastic properties of the physical GH hydrogel when compared to the covalent gelatin hydrogel (**sFig. 4g,h**)”. It should be noted though that the various domains of the bicontinuous hydrogels are not pure hydrogels of gelatin and guest host hydrogels, but rather a mixture of both. We have conducted additional studies via confocal imaging to better characterize the chemical compositions of these domains.

3. A very wide range of domain sizes in some pictures (see Fig. 3b) can be observed. Does the mixing intensity affect the bicontinuous morphology?

Response: Thank you for this question. Mixing intensity does affect bicontinuous morphology. When mixing the hydrogel, we add both solutions (as noted in **Fig. 1a**) to a blunt ended syringe (**Supplemental Movie 1**) and mix in circular motions before transferring the hydrogel into a mold for further gelation or into a syringe for injection. With minimal mixing of the hydrogel (two revolutions), the characteristic morphology of the bicontinuous hydrogel is seen in some regions but not others, and the GR and GP domains are larger than after further mixing (**sFig. 2a**). With additional mixing, the bicontinuous structure stabilizes and further mixing does not change this characteristic structure (10 revolutions versus 25 revolutions). For all studies reported in this manuscript, hydrogels were mixed above this threshold, ensuring a consistent structure. We have added an additional supplemental figure (**sFig. 2a**), supplemental movie (**Supplemental Movie 1**), and the following text to the manuscript to clarify this point, “These solutions were added to a blunt ended syringe and mixed with circular revolutions until the structure stabilized (>10 circular revolutions, **sFig. 2a, Supplementary Movie 1**)”.

4. The modulus in Fig. 2e can show some difference between the modulus at different points. However, it is a clear way to show the modulus difference in GR and GP phases. A modulus map superposed on the morphology can make the difference clearer.

Response: We thank the reviewer for this suggestion. The goal of introducing the AFM modulus map was to demonstrate the micron-scale mechanical heterogeneity present in the bicontinuous hydrogels. With this approach it was not technically possible to superimpose the structure with the mechanics. This has now been clarified with the following text, “To determine if the observed heterogeneity in structure (**sFig. 7a**) correlated with mechanical heterogeneity at the microscale, atomic force microscopy was used to generate force maps (105 x 105 μm) (**Fig. 2e**).”

To further investigate the mechanical properties of the different domains, we then performed AFM- nanoindentation and correlated these data with fluorescent domains based on overlapping fiduciary beads introduced to the hydrogel. The sequential acquisition of this data limits our ability to precisely colocalize the fluorescence precisely at each point but does enable us to make conclusions based on an aggregate of points. To better clarify this, we have reorganized the modulus maps in **Fig. 2e** and have reorganized **Supplementary Fig. 7b-d** to explain how we correlated fluorescence from confocal images with AFM-nanoindentation locations.

5. A control experiment using the pure GH physical gel for cell migration is necessary to prove that the rapid migration in the bicontinuous gel is not due to the low modulus of the GP phase.

Response: Thank you for this suggestion. In response, we have performed additional studies to measure cell migration through the GH physical gel alone. Very little/no migration was observed in the GH physical gel alone. We have added quantification and discussion of these findings to

the revised manuscript (**sFig. 12e,f**). Thus, we do not believe that the migration observed is due to the low modulus of the domain.

6. Although it is stated that cells use the interfaces between the GR and GP domains to migrate, direct observation in Fig. 3 is somewhat difficult. Moreover, such a statement implies that the interfacial area between two domains is readily broken by the cell, i.e., the interfacial strength is low, which can be evaluated using the bilayer structure in Fig. 5. The difference in the outgrowth in Fig. 5c using guest-host gel and agarose gel indicates the possible difference in the interfacial strength.

Response: We appreciate the opportunity to discuss this further. To start, we have added a zoomed-in insert of cells that have crawled along a GR domain to **Fig. 3b** to better visualize cell movement through the bicontinuous hydrogel. We agree that the malleability of the different phases that compose the interface may play a role in how easily a cell is able to migrate. Not only is this observed in **Fig. 5c** with the differences between GH with agarose, but it can also be observed when comparing 3D interdigitating interfaces of GH and agarose with gelatin, as observed in the difference between the absolute values of cell migration in the bicontinuous hydrogel (GH, **Fig. 3d**) versus the granular hydrogel composites (**Fig. 5h**). Similarly, we also found that increasing the mechanical properties of the agarose in the bilayer system did not alter the extent of migration (**sFig. 14b-d**). This may suggest that chemical composition plays a larger role in dictating this interfacial strength than the mechanical properties or that the range of materials probed were not sufficient to elicit variation in cellular behavior.

We have added the following additional text to address this point into the manuscript, “Enhanced migration due to bicontinuity was replicated in alternate material systems (planar interfaces and granular composites) and established that the microinterfaces, rather than the differential mechanics of each phase or their specific chemical composition, controlled this migration process (**Fig. 5**). Increased migration in GH-based interfaces when compared to agarose-based interfaces (**Fig. 5**) suggests that malleability of the interface constituents may alter the interfacial strength and ease of cell movement between domains, but further experimentation would be required to explore this hypothesis.”

7. In the bilayer experiments (Fig. 5), it is necessary to compare the cell outgrowth in the direction perpendicular to the interface. The conclusion is solid only when vertical outgrowth is much smaller than parallel.

Response: We agree and now provide this data in the revised manuscript (**sFig14. b,f**). Briefly, the top-down maximum projections originally provided made it difficult to assess the extent of ingrowth into the gelatin layer with this system. To explore this, we performed additional analysis on the spheroid outgrowth studies reported in **Fig. 5a-d** and **sFig. 14** in Imaris, in which we combined spheroid area into a single object and measured the outgrowth perpendicular to the interface. This analysis is slightly different from the other data presented, as it was not possible to subtract individual spheroid radii. As a result, **sFig. 14b,f** of cell ingrowth into the gelatin layer includes spheroid radius into the gelatin layer (i.e., “Nothing” shows a value > 0).

Cell outgrowth in the perpendicular direction is minimal, with levels ~5x lower than the outgrowth along the interface between the two layers. We have added the following text to the manuscript, “Importantly, minimal outgrowth was observed perpendicular to the interface into the bottom gelatin layer no matter the composition of the top layer (**sFig 14a,b**).” and “Again, minimal outgrowth perpendicular to the interface was observed (**sFig. 14f**).” To further visualize this differential migratory propensity, we have also added Imaris side-views of reconstructions (**sFig. 14a**) that demonstrate that cell outgrowth occurs largely along the interface between gelatin and the other material and that ingrowth into the gelatin layer is minimal.

Reviewer #2

The work by Xu et al. presents a design of non-miscible biopolymer to obtain discrete, yet continuous microstructural features in one mixed hydrogel. This biopolymer is applied toward the study of cell migration in the context of possible regenerative medicine application. The resulting microinterface promote cell migration along its boundary. They proceed to demonstrate that the material has pro-migratory properties in vivo. This is certainly a highly relevant way to achieve cell migration with contact guidance using material formulation. There are, however, a number of issues that need to be addressed.

Main concerns:

1-There is currently no real discussion apart from a few experimental justifications provided in the results section. The authors need to better situate their findings and the relevance of this material in light of the advances in the field and to provide context to non-experts. The provided outlook is certainly not sufficient.

Response: We appreciate this feedback and have updated the text with a new Discussion section to better situate our findings with respect to the field and innovations we believe are encompassed in this work. This was previously limited by original manuscript length restrictions, which can be expanded in this revision.

2- The stiffness measurement maps in Fig 2 are difficult to interpret.

Response: Please see response to Reviewer 1 (comment 4) above, who raised a similar point. We have reorganized and reformatted Fig. 2 to improve clarity.

3- I am quite confused by the diverging results for the mechanical test. On one hand, the storage modulus remains the same for all formulation while the loss modulus increased with %. On the other hand, average AFM measurements indicate that the modulus decrease with %. I would expect the two measurement methods to be somewhat correlated. Is there something I am missing here?

Response: This is an important point. In general, while both AFM-nanoindentation and rheology evaluate the mechanics of a material, they do so at very different length scales — with AFM-nanoindentation probing at the μm scale and rheology probing the bulk material. Similarly, heterogeneity in AFM-nanoindentation measurements may occur based on the location at which the material is probed. For example, if probing within the GR domains (which would likely be the lower bound of the 0% hydrogel), the stiffnesses will be higher compared to probing within the GP domains (**Fig. 2g**). Another important aspect is the nature of the applied deformation. Given the constituents of our bicontinuous hydrogel, the frequency at which the material is probed will have an impact on the mechanical properties obtained. For example, in **sFig. 4b**, we highlight the variations in the bulk mechanical properties of our hydrogels when probed at different frequencies. At higher frequencies, the 3% hydrogels trend toward higher storage moduli, whereas at lower frequencies, the 3% hydrogels trend toward lower storage moduli when compared to the 0% hydrogels. The combination of the rheologic studies with the AFM-nanoindentation may suggest that our hydrogels were indented at lower frequencies than the 1 Hz frequency at which we perform our rheologic studies in **Fig. 1c,e**.

We have added the following text to the manuscript to clarify these points “With this biopolymer-based bicontinuous hydrogel, we demonstrated that mixing and stabilization of immiscible phases could be conducted under cytocompatible conditions to engender distinct domains with high interfacial surface area and connectivity and bulk properties that changed with the introduction of the physical network (**Fig. 1**). Interestingly, we noted a decrease in the Young’s modulus for our bicontinuous hydrogels with AFM-nanoindentation when compared to the uniform

hydrogel, which may be due to the differences in both the spatial scale and frequency of indentation at which the hydrogels were probed between these two mechanical tests (**Fig. 2**). “

4 – It appears that the AFM measurements were performed on the upper surface of these hydrogels. Is it possible that this hydrogel/ air or liquid interface could exhibit altered boundary condition / mechanical properties? It should be possible to process this kind of hydrogel for cryosection (maybe using two color OCT) and then perform AFM on the internal structure of the hydrogel.

Response: We thank the reviewer for raising this point. Unfortunately, cryosectioning introduces artifacts due to ice crystal formation (which we learned by attempting the suggested studies). Hydrogels, as soft substrates with high water content, are particularly susceptible to changes in structure and mechanical properties from this processing.¹ Thus, to address this point, we fabricated a hydrogel (1% bicontinuous hydrogel), allowed it to undergo gelation, and then cut the hydrogel with a razor blade. We probed the interior surface of this hydrogel and compared this to our previous AFM-nanoindentation results for the outer surface and found no significant difference in the mechanical properties in this hydrogel (**sFig. 7e**). We've added the following associated text, “Furthermore, mechanical properties were equivalent on external (i.e., surface) and internal surfaces of the hydrogel (**sFig. 7e**), confirming that surface-level mechanics are representative of internal properties as well.”

5- The claim that bicontinuous hydrogels may provide energetically favorable paths for cell migration is a bit bold given that there is no direct evidence provided for this.

Response: We agree that this statement was speculative, and that further experimentation would be needed to prove this. We have reworded the text as follows: “These findings suggest that the high surface area and microinterfaces of the bicontinuous hydrogels may provide favorable paths for cell migration.”

6- Given the impact of the MMP inhibitor, do the authors know what is the actual porosity of the GH side?

Response: This is an interesting question, and one that would certainly prove useful for understanding other biophysical features of the hydrogel that may be supporting the cell migration trends observed. Hydrogels are typically considered to have nano-scale porosity, based on the mesh size, or area between adjacent crosslinks in the polymer network.² This is why, without porogens or other degradable features³, most bulk hydrogels are limited with regards to cell migration. For free migration to occur, pore sizes on the micron scale are required, and these would be detectable by confocal microscopy. We observed no such micron-scale pores in our highest resolution imaging.

To better understand the spatial relationship between the various polymers in our hydrogel and to detect if certain regions were more porous, we individually labelled different polymers within the hydrogel to understand their interactions with each other (**sFig. 2b**). Based on this study, we believe that the GP domain is comprised of the GH material (predominantly the CD-HA) with lower concentrations of the other bicontinuous hydrogel components (e.g., AD-HA, gelatin). To clarify this, we have added the following text to the manuscript, “We observed with a 3% GH formulation (**sFig. 2b**) that polymers colocalize based on the original solutions used in hydrogel formation (**Fig. 1a**). That is, gelatin and AD-HA (guest) were the primary components of the GR domain, while CD-HA (host) was the main constituent of the GP domain. Given that transglutaminase (which was originally in the CD-HA solution) must interact with gelatin to crosslink that domain (**Fig. 1c**), all components of the two solutions are likely diffusing and interacting with each other within and across the domains to varying extents. These findings suggest that the strong physical interactions between the components of the two solutions result in an immiscibility that stabilizes

the interfaces into bicontinuous structures, where each domain consists of a unique hydrogel composition.”

7- The use of the RGD peptide here is surprising. Wouldn't it be better to use the GFOGER motif instead since gelatin and not a fibrinogen is used in the system?

Response: We thank the reviewer for raising this point. It is true that GFOGER is a well-known sequence on collagen that supports cell adhesion, whereas RGD is a peptide classically associated with fibronectin, another common ECM protein. However, our goal in this study was to identify a small molecule that could penetrate through the 3D hydrogel structure to serve as a competitive inhibitor to integrin-mediated binding, as we have done in the past.⁴ Additionally, while gelatin is a derivative of collagen, the change in conformation and chemical identity of gelatin from collagen influences the availability of GFOGER versus RGD. Specifically, collagen's native structure confines RGD motifs that are freed once gelatin adopts its random structure.⁵ As a result, RGD seems to be the predominant binding motif in gelatin, despite gelatin having conserved peptide sequences with collagen⁶. To clarify this, we have added the following text: “To probe the role of these adhesion sites on migration, we introduced soluble RGD as a competitive inhibitor of integrin binding, due to the presence of RGD moieties on gelatin that are exposed due to collagen denaturing. ⁶”

8- The explant and in vivo experiments are quite interesting and relevant and show that cells will readily invade within the hydrogel. However, it is unclear which kind of cells actually enter the hydrogel, particularly in the in vivo experiment. Are these chondrocytes, immune cells or something else?

Response: We agree that the nuclear staining does not indicate what kind of cells have entered the hydrogel. To minimize the ingrowth from exogenous rat cells, we specifically cut our resected explants into 1 mm sections and analyzed hydrogel remaining within the defect in only the most central middle sections (**Fig. 6d**), in which the closest cell source are the bovine meniscus cells. Unfortunately, meniscus cells lack specific surface markers with which to identify if cells are truly from the meniscus.⁷ To determine if cells are of immune origin, we labeled our in vivo samples (from **Fig. 6d-f**) with CD68, a pan-macrophage marker (**sFig. 19**).⁸ We found that less than 5% of the cells that infiltrated into our hydrogel were positive for CD68, suggesting that there are few macrophages, and likely few immune cells that have infiltrated to the middle regions of the explants. Given this finding, we would expect that most infiltrating cells are of meniscus origin. The following additional text has been added regarding this point, “Further labeling suggested that infiltration from immune cells was limited, as only a small population of infiltrated cells exhibited CD68 staining, a macrophage marker (**sFig. 19**).”

9- For those two experiments, is the bicontinuous hydrogel more efficient at driving cell migration compared to other uniform hydrogels (collagen for instance)? Performing such an experiment would provide a good comparison point for the bicontinuous material.

Response: We thank the reviewer for suggesting an expansion of the comparisons of the bicontinuous hydrogel to other uniform materials. We compared our bicontinuous hydrogels with other uniform materials, including previously used concentrations of collagen⁹, to assess cell outgrowth from explants using the same experimental setup found from **Fig. 6a**. Given collagen's fibrillar topology and ability to be remodeled and degraded¹⁰, we also performed these studies on a uniform synthetic hydrogel (5wt% acrylated hyaluronic acid)¹¹. When comparing the values for infiltration into these materials (50 μm) to the ones obtained in our bicontinuous hydrogels (~150 μm , **Fig. 6c**), we note an approximately 3-fold increase in the maximum cell infiltration with the bicontinuous materials (**sFig. 16**, left panel). Similarly, most cells were found on the surface of the uniform materials, compared to our bicontinuous hydrogels, where greater than 30% were found within the hydrogel (**sFig. 16**, right panel). We have added the following text, “We found similar

trends between 0% GH hydrogel and other uniform hydrogels, including collagen and acrylated hyaluronic acid (**sFig. 1**), which all demonstrated minimal cell infiltration when compared to the bicontinuous hydrogels (**sFig. 16**)."

10- For the in vivo conditions, how much of the material is still intact at the end of the experiment? Are the invading cells replacing some of the material with native ECM?

Response: We thank the reviewer for this comment. For the material to be used as a tissue engineered/regenerative product, the material should be replaced over time by native ECM to support tissue repair. Thus, to address this question, we labeled our fluorescent hydrogel by incorporating FITC-labeled gelatin and assessed the % of the defect space that did not contain fluorescence. There was no statistically significant difference in the % areas measured in this experiment. Thus, we included the following text: "In monitoring the fluorescent signal from incorporated gelatin, there was similar fluorescence observed in all samples after 2 weeks (**sFig. 20**), indicating that remodeling in vivo was similar across formulations with respect to gelatin degradation." We would expect that cell infiltration into these regions would result in ECM deposition, but it has not been possible to measure this in our samples.

Minor concern:

1- Appreciating how the micro-interface support cell migration is difficult with the current pictures in the different figures. Showing higher magnification pictures (e.g. insert) of the migrating cells would be helpful here.

Response: We thank the reviewer for pointing this out. In **Fig. 2b**, we have added a zoom-in inset highlighting cells at the interface of the GP domain and larger GR domain in the representative example.

2- Is it possible to obtain a similar bicontinuous effect if gelatin is replaced with full-length collagen?

Response: To determine if bicontinuous structures can be induced with collagen, we mixed collagen and our physical GH network together, in which one of the solutions consisted of collagen and AD-HA and the other solution consisted of CD-HA and the neutralization solution (for collagen). In **sFig. 21**, we see that the structure of 6wt% collagen¹² is relatively homogenous (likely given the higher wt%), whereas 6wt% collagen in the presence of our physical network exhibits slight structural heterogeneity. However, this structure differs from the clear bicontinuous structure seen in our hydrogel. This may be due to a number of differences between collagen and gelatin including intrinsic physical interactions, crosslinking mechanism, stable wt%, and structure. The following text has been added to the manuscript discussion, "Experimentation with constituent ratios and immiscibility may be required, as simply adding two viscous mixtures (e.g., collagen, AD-HA and CD-HA) was insufficient to obtain a bicontinuous structure (**sFig. 21**)."

Reviewer #3

In this manuscript, Xu and co-authors describe the fabrication of a hydrogel by mixing hyaluronic acid (HA)-cyclodextrin (CD) and hyaluronic acid (HA)-adamantane (AD) with highly crosslinked gelatin. The authors claim the formation of a bicontinuous hydrogel and demonstrate cell migration behavior into this unique structure. The manuscript is well-written and represents a thoughtful experimental design. This manuscript is a strong candidate for publication in Nature Communications once the concerns raised by the reviewers are addressed by the authors.

1. It appears that the bicontinuous feature, which arises from the combination of HA-AD, HA-CD, and gelatin, is a defining characteristic of the material. Therefore, it would be appropriate to

include this information in the title and the abstract. Mentioning only gelatin and HA (instead of HA-AD and HA-CD and crosslinked gelatin) in the abstract may be somewhat misleading. Response: Thank you for this suggestion. We have altered our abstract text to highlight the specific role of the AD, CD, and transglutaminase in our bicontinuous hydrogel, as suggested by the reviewer. Regarding the title, we have added “biopolymer-based” to better specify the materials used, but it would be difficult to clarify guest-host and transglutaminase in the title. We hope that these updates together help to clarify the system.

2. The authors mixed transglutaminase and hyaluronic acid (HA)-cyclodextrin (CD) with hyaluronic acid (HA)-adamantane (AD) and gelatin, claiming the formation of a bicontinuous hydrogel due to limited miscibility of AD and CD. While it's clear that interactions between HA-AD and HA-CD resulted in phase separation (as demonstrated in extended Fig. 2c), it's essential to determine if there are any additional phases beyond what is observed through confocal imaging. What evidence supports the conclusion that HA-CD and HA-AD together form only one continuous phase?

Response: Thank you for this question, one that we ourselves have thought through extensively. Given that the material structures were present from the beginning (**Supplementary Movie 2**), we believe that the components of the hydrogel exhibit arrested immiscibility, in which both the AD-HA and CD-HA are required for this arrest, but do not necessarily form the GP domain itself. To support this, we completed labeling studies in which we fluorescently colocalized the different polymers in our system (**sFig. 2b**). Please see response to Reviewer 2, Comment 6 above where we detail this experiment and the additional text and supplementary figure on these findings. We did attempt alternate methods (such as FTIR analysis of the gels); however, processing of the samples for analysis disrupted the structure. Thus, we believe that this additional confocal imaging best characterizes the hydrogel structure and composition, since it can be maintained in the hydrated state.

3. The authors used a relatively high transglutaminase concentration (1U/mL) with 5 wt% gelatin for cross-linking. This may result in a densely crosslinked gelatin network, making it challenging for cells to migrate. It seems that using a lower transglutaminase concentration might yield a different outcome, potentially allowing cells to migrate through the gelatin-rich region as well. Did the authors experiment with different transglutaminase concentrations to create the bicontinuous hydrogel?

Response: We thank the reviewer for their insightful question. We actually began these studies by exploring even higher concentrations of transglutaminase (**sFig. 3**), based on prior literature.¹³⁻¹⁵ We found that these variations in the enzymatic concentration did not affect mechanical properties (G' , G'' , or $\tan(\delta)$), but did influence the amount of time that it took for the hydrogel to crosslink (quantified by a plateau in the mechanical properties). As such, we chose the lowest enzyme concentration of those we tested to enable enough time for the user to mix and transfer the material into the mold.

To confirm that a lower transglutaminase concentration would yield similar findings, we performed our studies in the 3% hydrogel, but used 0.5 U/mL of transglutaminase instead of 1 U/mL (**sFig. 6**). We saw a slight increase in the extent of cell migration within 3 days but observed that cells remained on the interfaces between GR and GP domains. These results suggest that cells continue to rely on interfaces for migration in the bicontinuous hydrogels, no matter the enzymatic crosslinking. We've added the following text to the manuscript to underscore this point, “Decreasing in enzymatic crosslinker concentration similarly supported cell migration along interfaces in the 3% bicontinuous hydrogel (**sFig. 8**).”

4. Fig. 6e shows extensive material degradation for 3%, along with a notably high cellularity (also evident from supplemental Fig. 7). These cells could potentially be different types of immune cells

infiltrating the site of material degradation, triggering inflammatory reactions. Have the authors identified the type of cells that infiltrated the defect site? Simple histological analysis could provide insight into the inflammatory environment. Given this situation, the bicontinuous hydrogel may not be suitable for in vivo applications.

Response: We thank the reviewer for raising this point. Please refer to response to Reviewer 2 Comment 8 who asked a similar question. Our new results, coupled with our additional data demonstrating degradation of the bicontinuous hydrogel over days (**sFig. 6**), suggests that the bicontinuous hydrogel is a suitable in vivo tissue repair strategy. Future long terms studies will of course be necessary to fully evaluate if this new material in the contexts of specific tissue engineering approaches.

5. The bicontinuous hydrogel appears to have low mechanical strength. The authors did not provide any degradation test results (both in vitro and in vivo). The evidence from the in vivo results presented in Fig. 7 suggests that this hydrogel may be highly unstable. It would be valuable for the authors to discuss the potential applications of this hydrogel in tissue engineering considering its mechanical properties and instability.

Response: The authors thank the reviewer for this great discussion point and the identification of further experimentation that would highlight the strength of our hydrogel. To address these studies, we performed degradation studies, both by measuring volume loss over time as well as uronic acid released (**sFig. 6b**).¹⁶ These results suggest that over the course of 8 days, 80% of the gel is degraded, with slowed continuous degradation afterwards. We've added the following additional text, "Importantly, the hydrogels were also structurally and chemically stable for at least a week (**sFig. 6b**)."

In musculoskeletal tissue engineering, two strategies may be undertaken to support tissue defects and the optimal one often depends on the type of defect that must be repaired. The first is to build robust biomaterials that match the mechanical properties of the surrounding tissue, such that the biomaterials may be inserted into defects or replace native tissue (e.g., hip implants) to assist in load-bearing. This strategy is often undertaken with large irreparable defects. The second is to introduce a cell-compatible, rapidly degradable material that is weak enough such that cells can rapidly migrate into the hydrogel, remodel their surrounding tissue, and degrade the substrate while depositing their own native ECM. This strategy is often more conducive to discrete and smaller defect spaces. In the latter example, a substrate that is conducive to these cellular interactions often must be of low mechanical strength and support tissue remodeling and prompt deposition of new tissue. This strategy underlies our bicontinuous material, which provides for ready space filling and rapid cell colonization for tissue repair applications.

6. Do the authors anticipate any potential limitations in the fabrication of a bicontinuous hydrogel using alternative materials? What challenges might other researchers encounter in their efforts to develop a bicontinuous hydrogel?

Response: We thank the reviewer for this question. We had conducted prior studies related to the formation of bicontinuous hydrogels with granular systems and have performed new studies related to forming these with alternate materials (e.g., collagen). To address this comment and elaborate on this point, we have changed our outlook to a discussion and explored the potential ease and application of other biopolymer-based bicontinuous hydrogels. Specifically, we've added the following text, "Our bicontinuous hydrogels rely on controlled immiscibility to form their characteristic interconnected structures. These structures likely arise due to the strong physical interactions between our guest and host moieties that served to arrest the interfaces temporarily, during which time the enzymatic crosslinking stabilizes these microdomains. To extend this strategy, other bicontinuous hydrogels could be fabricated using two immiscible constituents or employing phase separation (perhaps by engineering strong physical interactions or using

inherently immiscible high molecular weight polymers) and utilizing an additional stabilization strategy (such as photoinduced crosslinking).⁶⁰ Similarly, given the role of biopolymers in other block copolymer assemblies⁶¹ for cell-material interaction studies, introducing bicontinuous hydrogels using biopolymer-based block copolymers may also provide a viable option. Experimentation with constituent ratios and immiscibility may be required, however, as simply adding two viscous mixtures (e.g., collagen, AD-HA and CD-HA) was insufficient to obtain a bicontinuous structure (**sFig. 21**).

Enhanced migration due to bicontinuity was replicated in alternate material systems (planer interfaces and granular composites) and established that the microinterfaces, rather than the differential mechanics of each phase or their specific chemical composition, controlled this migration process (**Fig. 5**). Increased migration in GH-based interfaces, when compared to agarose-based interfaces (**Fig. 5**), suggests that malleability of the interface constituents may alter the interfacial strength and ease of cell movement between domains, but further experimentation would be required to explore this hypothesis. Additionally, attempting to tune features such as scale and tortuosity of the interconnected domains, will require further refinements.”

References

- 1 Ruan, J. L. *et al.* An improved cryosection method for polyethylene glycol hydrogels used in tissue engineering. *Tissue Eng Part C Methods* **19**, 794-801 (2013). <https://doi.org/10.1089/ten.TEC.2012.0460>
- 2 Rehmann, M. S. *et al.* Tuning and Predicting Mesh Size and Protein Release from Step Growth Hydrogels. *Biomacromolecules* **18**, 3131-3142 (2017). <https://doi.org/10.1021/acs.biomac.7b00781>
- 3 Annabi, N. *et al.* Controlling the porosity and microarchitecture of hydrogels for tissue engineering. *Tissue Eng Part B Rev* **16**, 371-383 (2010). <https://doi.org/10.1089/ten.TEB.2009.0639>
- 4 Loebel, C., Mauck, R. L. & Burdick, J. A. Local nascent protein deposition and remodelling guide mesenchymal stromal cell mechanosensing and fate in three-dimensional hydrogels. *Nature Materials* **18**, 883-891 (2019). <https://doi.org/10.1038/s41563-019-0307-6>
- 5 Davidenko, N. *et al.* Evaluation of cell binding to collagen and gelatin: a study of the effect of 2D and 3D architecture and surface chemistry. *J Mater Sci Mater Med* **27**, 148 (2016). <https://doi.org/10.1007/s10856-016-5763-9>
- 6 Liu, D., Nikoo, M., Boran, G., Zhou, P. & Regenstein, J. M. Collagen and Gelatin. *Annual Review of Food Science and Technology* **6**, 527-557 (2015). <https://doi.org/10.1146/annurev-food-031414-111800>
- 7 Verdonk, P. C. M. *et al.* Characterisation of human knee meniscus cell phenotype. *Osteoarthritis and Cartilage* **13**, 548-560 (2005). <https://doi.org/10.1016/j.joca.2005.01.010>
- 8 Qazi, T. H. *et al.* Anisotropic Rod-Shaped Particles Influence Injectable Granular Hydrogel Properties and Cell Invasion. *Advanced Materials* **34**, 2109194 (2022). <https://doi.org/10.1002/adma.202109194>
- 9 Baker, B. M. *et al.* Cell-mediated fibre recruitment drives extracellular matrix mechanosensing in engineered fibrillar microenvironments. *Nature Materials* **14**, 1262-1268 (2015). <https://doi.org/10.1038/nmat4444>
- 10 Jagiełło, A., Castillo, U. & Botvinick, E. Cell mediated remodeling of stiffness matched collagen and fibrin scaffolds. *Scientific Reports* **12**, 11736 (2022). <https://doi.org/10.1038/s41598-022-14953-w>

- 11 Shen, Y.-I. *et al.* Hyaluronic acid hydrogel stiffness and oxygen tension affect cancer cell fate and endothelial sprouting. *Biomaterials Science* **2**, 655-665 (2014). <https://doi.org:10.1039/C3BM60274E>
- 12 Davidson, M. D. *et al.* Programmable and contractile materials through cell encapsulation in fibrous hydrogel assemblies. *Science Advances* **7**, eabi8157 (2021). <https://doi.org:doi:10.1126/sciadv.abi8157>
- 13 Yang, G. *et al.* Enzymatically crosslinked gelatin hydrogel promotes the proliferation of adipose tissue-derived stromal cells. *PeerJ* **4**, e2497 (2016). <https://doi.org:10.7717/peerj.2497>
- 14 Xu, J. *et al.* Effect of transglutaminase crosslinking on the structural, physicochemical, functional, and emulsion stabilization properties of three types of gelatins. *LWT* **163**, 113543 (2022). <https://doi.org:https://doi.org/10.1016/j.lwt.2022.113543>
- 15 Zhou, M., Lee, B. H., Tan, Y. J. & Tan, L. P. Microbial transglutaminase induced controlled crosslinking of gelatin methacryloyl to tailor rheological properties for 3D printing. *Biofabrication* **11** (2019).
- 16 Cesaretti, M., Luppi, E., Maccari, F. & Volpi, N. A 96-well assay for uronic acid carbazole reaction. *Carbohydrate Polymers* **54**, 59-61 (2003). [https://doi.org:https://doi.org/10.1016/S0144-8617\(03\)00144-9](https://doi.org:https://doi.org/10.1016/S0144-8617(03)00144-9)

REVIEWERS' COMMENTS

Reviewer #1 (Remarks to the Author):

The authors gave a clear response to my questions and revised the manuscript accordingly. I suggest publication in the present form.

Reviewer #2 (Remarks to the Author):

This is a revised version of manuscript by Xu et al. The authors have made major efforts to address the concerns raised initially. The additions to the text and new data help in clarifying several aspects that were originally unclear. It is also appreciated that they tried to perform experiments that are technically challenging. The results provided really highlights how to take advantage of such materials. The improved discussion also put more context in the importance of this work. I believe this brings some really interesting and overdue insights to the field.

On a side note, the explanation provided for the change in material properties between the AFM and rheological measurements in the discussion is difficult to understand. I agree that the observed effect is likely related to the scale of the measurement and frequency dependency, but the explanation is somewhat confusing in the text and seems to be missing something.

Apart from this small detail, my concerns have been well addressed. I congratulate the authors on this nice piece of work.

Reviewer #3 (Remarks to the Author):

The authors have addressed all comments raised by the reviewers, thereby significantly improving the quality of the manuscript. This reviewer does not have any further comments for the authors. The current version of the manuscript is now deemed suitable for publication in Nature Communications.

Dear Editor and Reviewers,

We are grateful for the additional review of our revised work. Based on this feedback, we have modified the manuscript. Point-by-point responses to each reviewer are provided below in addition to the requested documents.

Reviewer #1 (Remarks to the Author):

The authors gave a clear response to my questions and revised the manuscript accordingly. I suggest publication in the present form.

Response: We thank the reviewer for their previous comments and for the favorable suggestion.

Reviewer #2 (Remarks to the Author):

This is a revised version of manuscript by Xu et al. The authors have made major efforts to address the concerns raised initially. The additions to the text and new data help in clarifying several aspects that were originally unclear. It is also appreciated that they tried to perform experiments that are technically challenging. The results provided really highlights how to take advantage of such materials. The improved discussion also put more context in the importance of this work. I believe this brings some really interesting and overdue insights to the field. On a side note, the explanation provided for the change in material properties between the AFM and rheological measurements in the discussion is difficult to understand. I agree that the observed effect is likely related to the scale of the measurement and frequency dependency, but the explanation is somewhat confusing in the text and seems to be missing something. Apart from this small detail, my concerns have been well addressed. I congratulate the authors on this nice piece of work.

Response: We thank the reviewer for their added time in reviewing the revised manuscript and appreciate the acknowledgment of the technical difficulty of the added experiments. The reviewer was correct that a phrase was missing from the discussion section regarding the AFM and rheology, and we appreciate being notified of the mistake. The authors have reworded the text to be the following: “Interestingly, with increasing bicontinuity we noted a decrease in the Young’s modulus with AFM-nanoindentation, but equivalent storage moduli with rheology. This discrepancy may be due to the differences in both the spatial scale and frequency of indentation at which the hydrogels were probed between these two mechanical tests (Fig. 2).”

Reviewer #3 (Remarks to the Author):

The authors have addressed all comments raised by the reviewers, thereby significantly improving the quality of the manuscript. This reviewer does not have any further comments for the authors. The current version of the manuscript is now deemed suitable for publication in Nature Communications.

Response: We thank the reviewer for their previous suggestions which have strengthened the publication considerably.